# Boosting the Transferability of Adversarial Attacks with Reverse Adversarial Perturbation

**Zeyu Qin[1*], Yanbo Fan[2*], Yi Liu[1], Li Shen[3], Yong Zhang[2], Jue Wang[2], Baoyuan Wu[1†]**
[1]School of Data Science, Shenzhen Research Institute of Big Data,
The Chinese University of Hong Kong, Shenzhen
[2]Tencent AI Lab
[3]JD Explore Academy
{zeyu6181136, fanyanbo0124, yiliuhk2000}@gmail.com
{mathshenli, zhangyong201303, arphid}@gmail.com
wubaoyuan@cuhk.edu.cn

## Abstract

Deep neural networks (DNNs) have been shown to be vulnerable to adversarial examples, which can produce erroneous predictions by injecting imperceptible perturbations. In this work, we study the transferability of adversarial examples, which is significant due to its threat to real-world applications where model architecture or parameters are usually unknown. Many existing works reveal that the adversarial examples are likely to overfit the surrogate model that they are generated from, limiting its transfer attack performance against different target models. To mitigate the overfitting of the surrogate model, we propose a novel attack method, dubbed *reverse adversarial perturbation* (RAP). Specifically, instead of minimizing the loss of a single adversarial point, we advocate seeking adversarial example located at a region with unified low loss value, by injecting the worst-case perturbation (*i.e.*, the reverse adversarial perturbation) for each step of the optimization procedure. The adversarial attack with RAP is formulated as a min-max bi-level optimization problem. By integrating RAP into the iterative process for attacks, our method can find more stable adversarial examples which are less sensitive to the changes of decision boundary, mitigating the overfitting of the surrogate model. Comprehensive experimental comparisons demonstrate that RAP can significantly boost adversarial transferability. Furthermore, RAP can be naturally combined with many existing black-box attack techniques, to further boost the transferability. When attacking a real-world image recognition system, *i.e.*, Google Cloud Vision API, we obtain 22% performance improvement of targeted attacks over the compared method. Our codes are available at: https://github.com/SCLBD/Transfer_attack_RAP.

## 1 Introduction

Deep neural networks (DNNs) have been successfully applied in many safety-critical tasks, such as autonomous driving, face recognition and verification, *etc*. However, it has been shown that DNN models are vulnerable to adversarial examples [9, 12, 27, 32, 35, 44, 45, 50], which are indistinguishable from natural examples but make a model produce erroneous predictions. For real-world applications, the DNN models are often hidden from users. Therefore, the attackers need to generate the adversarial examples under black-box setting where they do not know any information of the target model [2, 3, 18, 32]. For black-box setting, the adversarial transferability matters since

*Equal contribution. †Corresponding author.
‡This work is done when Zeyu Qin is a research intern at Tencent AI Lab.

36th Conference on Neural Information Processing Systems (NeurIPS 2022).

it can allow the attackers to attack target models by using adversarial examples generated on the surrogate models. Therefore, learning how to generate adversarial examples with high transferability has gained more attentions in the literature [5, 10, 14, 23, 26, 38, 48].

Under white-box setting where the complete information of the attacked model (*e.g.*, architecture and parameters) is available, the gradient-based attacks such as PGD [27] have demonstrated good attack performance. However, they often exhibit the poor transferiability [5, 48], *i.e.*, the adversarial example $\boldsymbol{x}^{adv}$ generated from the surrogate model $\mathcal{M}^S$ performs poorly against different target models $\mathcal{M}^T$. The previous works attribute that to the overfitting of adversarial examples to the surrogate models [5, 24, 48]. Figure 1 (b) gives an illustration. The PGD attack aims to find an adversarial point $\boldsymbol{x}^{pgd}$ with minimal attack loss, while doesn't consider the attack loss of the neighborhood regions round $\boldsymbol{x}^{pgd}$. Due to the highly non-convex of deep models, when $\boldsymbol{x}^{pgd}$ locates at a sharply local minimum, a slight change on model parameters of $\mathcal{M}^S$ could cause a large increase of the attack loss, making $\boldsymbol{x}^{pgd}$ fail to attack the perturbed model.

Many techniques have been proposed to mitigate the overfitting and improve the transferability, including input transformation [6, 48], gradient calibration [14], feature-level attacks [17], and generative models [30], etc. However, there still exists a large gap of attack performance between the transfer setting and the ideal white-box setting, especially for targeted attack, requiring more efforts for boosting the transferability.

In this work, we propose a novel attack method called *reverse adversarial perturbation* (RAP) to alleviate the overfitting of the surrogate model and boost the transferability of adversarial examples. We encourage that $\boldsymbol{x}^{adv}$ is not only of low attack loss but also locates at a local flat region, *i.e.*, the points within the local neighborhood region around $\boldsymbol{x}^{adv}$ should also be of low loss values. Figure 1 (b) illustrates the difference between the sharp local minimum and flat local minimum. When the model parameter of $\mathcal{M}^S$ has some slight changes, the variation of the attack loss *w.r.t.* the flat local minimum is less than that of the sharp one. Therefore, the flat local minimum is less sensitive to the changes of decision boundary. To achieve this goal, we formulate a min-max bi-level optimization problem. The inner maximization aims to find the worst-case perturbation (*i.e.*, that with the largest attack loss, and this is why we call it reverse adversarial perturbation) within the local region around the current adversarial example, which can be solved by the projected gradient ascent algorithm. Then, the outer minimization will update the adversarial example to find a new point added with the provided reverse perturbation that leads to lower attack loss. Figure 1 (a) provides an illustration of the optimization process. For $t$-th iteration and $\boldsymbol{x}^t$, RAP first finds the point $\boldsymbol{x}^t + \boldsymbol{n}^{rap}$ with max attack loss within the neighborhood region of $\boldsymbol{x}^t$. Then it updates $\boldsymbol{x}^t$ with the gradient calculated by minimizing the attack loss *w.r.t.* $\boldsymbol{x}^t + \boldsymbol{n}^{rap}$. Compared to directly adopting the gradient at $\boldsymbol{x}^t$, RAP could help escape from the sharp local minimum and pursue a relatively flat local minimum. Besides, we design a late-start variant of RAP (RAP-LS) to further boost the attack effectiveness and efficiency, which doesn't insert the reverse perturbation into the optimization procedure in the early stage. Moreover, from the technical perspective, since the proposed RAP method only introduces one specially designed perturbation onto adversarial attacks, one notable advantage of RAP is that it can be naturally combined with many existing black-box attack techniques to further boost the transferability. For example, when combined with different input transformations (*e.g.*, the random resizing and padding in Diverse Input [48]), our RAP method consistently outperforms the counterparts by a clear margin.

Our main contributions are three-fold: **1)** Based on a novel perspective, the flatness of loss landscape for adversarial examples, we propose a novel adversarial attack method RAP that encourages both the adversarial example and its neighborhood region be of low loss value; **2)** we present a vigorous experimental study and show that RAP can significantly boost the adversarial transferability on both untargeted and targeted attacks for various networks also containing defense models; **3)** we demonstrate that RAP can be easily combined with existing transfer attack techniques and outperforms the state-of-the-art performance by a large margin.

## 2 Related Work

The black-box attacks can be categorized into two categories: 1) *query-based attacks* that conduct the attack based on the feedback of iterative queries to target models, and 2) *transfer attacks* that use the adversarial examples generated on some surrogate models to attack the target models. In this work,

we focus on the transfer attacks. For surrogate models, existing attack algorithms such as FGSM [12] and I-FGSM [21] could achieve good attack performance. However, they often overfit the surrogate models and thus exhibit poor transferability. Recently, many works have been proposed to generate more transferable adversarial examples [5, 6, 13, 14, 17, 19, 20, 24, 39, 40, 41, 43, 48], which we briefly summarize as below.

**Input transformation:** Data augmentation, which has been shown to be effective in improving model generalization, has also been studied to boost the adversarial transferability, such as randomly resizing and padding [48], randomly scaling [24], and adversarial mixup [41]. In addition, the work of Dong et al. [6] uses a set of translated images to compute gradient and get the better performance against defense models. Expectation of Transformation (EOT) method [1] synthesizes adversarial examples over a chosen distribution of transformations to enhance its adversarial transferability. **Gradient modification:** Instead of the I-FGSM, the work of Dong et al. [5] integrates momentum into the updating strategy. And Lin et al. [24] uses the Nesterov accelerated gradient to boost the transferability. The work of Wang and He [40] aims to find a more stable gradient direction by tuning the variance of each gradient step. There are also some model-specific designs to boost the adversarial transferability. For example, Wu et al. [43] found that the gradient of skip connections is more crucial to generate more transferable attacks. The work of Guo et al. [14] proposed LinBP to utilize more gradient of skip connections during the back-propagation. However, these methods tend to be specific to a particular model architecture, such as skip connection, and it is nontrivial to extend the findings to other architectures or modules. **Intermediate feature attack:** Meanwhile, Huang et al. [17], Inkawhich et al. [19, 20] proposed to exploit feature space constraints to generate more transferable attacks. Yet they need to identity the best performing intermediate layers or train one-vs-all binary classifies for all attacked classes. Recently, Zhao et al. [49] find iterative attacks with much more iterations and logit loss can achieve relatively high targeted transferability and exceed the feature-based attacks. **Generative models:** In addition, there have been some methods utilizing the generative models to generate the adversarial perturbations [28, 30, 31]. For example, the work of Naseer et al. [30] proposed to train a generative model to match the distributions of source and target class, so as to increase the targeted transferability. However, the learning of the perturbation generator is nontrivial, especially on large-scale datasets.

In summary, the current performance of transfer attacks is still unsatisfactory, especially for targeted attacks. In this work, we study adversarial transferability from the prespective of the flatness of adversarial examples. We find that adversarial examples located at flat local minimum will be more transferable than those at sharp local minimum and propose an novel algorithm to find adversarial example that locates at flat local minimum.

## 3 Methodology

### 3.1 Preliminaries of Transfer Adversarial Attack

Given an benign sample $(\boldsymbol{x}, y) \in (\mathcal{X}, \mathcal{Y})$, the procedure of transfer adversarial attack is firstly constructing the adversarial example $\boldsymbol{x}^{adv}$ within the neighborhood region $\mathcal{B}_\epsilon(\boldsymbol{x}) = \{\boldsymbol{x}' : \|\boldsymbol{x}' - \boldsymbol{x}\|_p \leq \epsilon\}$ by attacking the white-box surrogate model $\mathcal{M}^s(\boldsymbol{x}; \boldsymbol{\theta}) : \mathcal{X} \to \mathcal{Y}$, then transferring $\boldsymbol{x}^{adv}$ to directly attack the black-box target model $\mathcal{M}^t(\boldsymbol{x}; \boldsymbol{\phi}) : \mathcal{X} \to \mathcal{Y}$. The attack goal is to mislead the target model, *i.e.*, $\mathcal{M}^t(\boldsymbol{x}^{adv}; \boldsymbol{\phi}) \neq y$ (untargeted attack), or $\mathcal{M}^t(\boldsymbol{x}^{adv}; \boldsymbol{\phi}) = y_t$ (targeted attack) with $y_t \in \mathcal{Y}$ indicting the target label. Taking the target attack as example, the general formulation of many existing transfer attack methods can be written as follows:

$$\min_{\boldsymbol{x}^{adv} \in \mathcal{B}_\epsilon(\boldsymbol{x})} \mathcal{L}(\mathcal{M}^s(\mathcal{G}(\boldsymbol{x}^{adv}); \boldsymbol{\theta}), y_t). \tag{1}$$

The loss function $\mathcal{L}$ is often set as the cross entropy (CE) loss [48] or the logit loss [49], which will be specified in later experiments. Besides, the formulation of untargeted attack can be easily obtained by replacing the loss function $\mathcal{L}$ and $y_t$ by $-\mathcal{L}$ and $y$, respectively.

Since $\mathcal{M}^s$ is white-box, if $\mathcal{G}(\cdot)$ is set as the identity function, then any off-the-shelf white-box adversarial attack method can be adopted to solve Problem (1), such as I-FSGM [21], MI-FGSM [5], *etc*. Meanwhile, existing works have designed different $\mathcal{G}(\cdot)$ functions and developed the corresponding optimization algorithms, to boost the adversarial transferability between surrogate and target models. For example, $\mathcal{G}(\cdot)$ is specified as random resizing and padding (DI) [48], translation transformation (TI) [6], scale transformation (SI) [24], and adversarial mixup (Admix) [41].

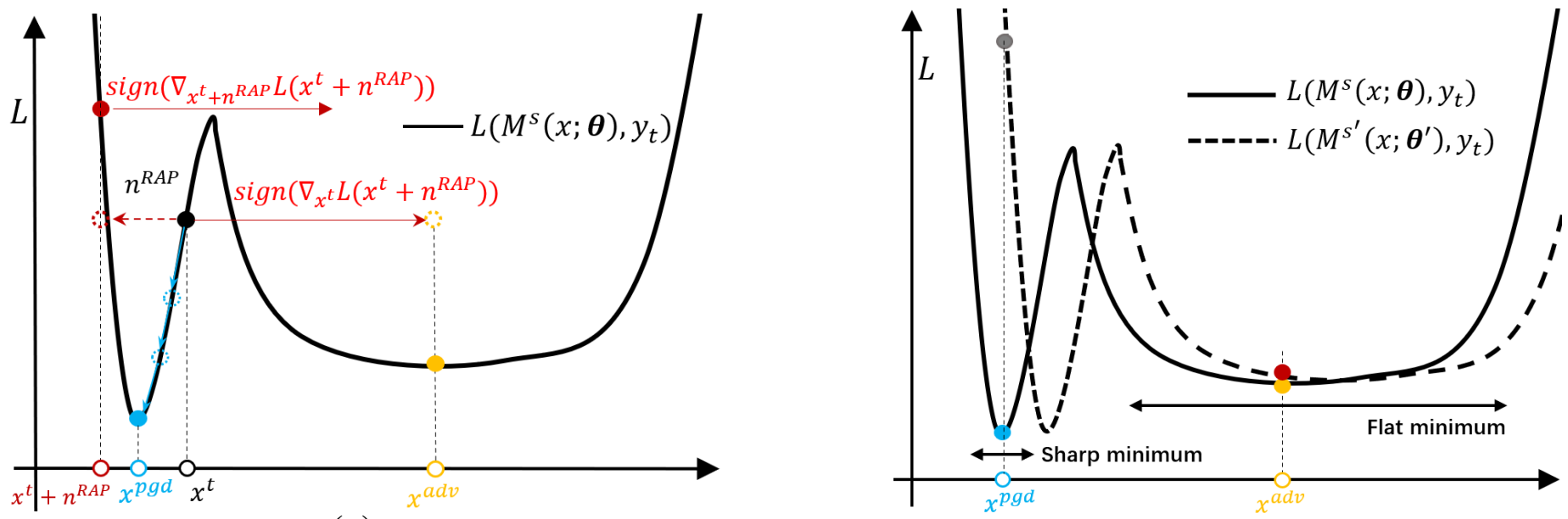

(a) (b)

Figure 1: These two plots are schematic diagrams in 1D space. The x-axis means the value of input $\boldsymbol{x}$. The y-axis means the value of attack loss function $\mathcal{L}$. (a) Illustration of our attack method and the original PGD attack. (b) Illustration of attack loss landscape of $\mathcal{M}^S$ and $\mathcal{M}^{S'}$. $\mathcal{M}^{S'}$ denotes a slight change on the model parameters of $\mathcal{M}^S$. The blue and yellow dots correspond to attacks located at different local minima on $\mathcal{M}^S$, respectively. The gray and red points are their counterparts on $\mathcal{M}^{S'}$.

### 3.2 Reverse Adversarial Perturbation

As discussed above, although having good performance for the white-box setting, the adversarial examples generated from $\mathcal{M}^S$ exist poor adversarial transferability on $\mathcal{M}^T$, especially for targeted attacks. The previous works attribute this issue to the overfitting of adversarial attack to $\mathcal{M}^S$ [4, 5, 6, 38, 46]. As shown in Figure 1 (b), when $\boldsymbol{x}^{pgd}$ locates at a sharp local minimum, it is not stable and is sensitive to changes of $\mathcal{M}^S$. When having some changes on model parameters, $\boldsymbol{x}^{pgd}$ could results in a high attack loss against $\mathcal{M}^{S'}$ and lead to a failure attack.

To mitigating the overfitting to $\mathcal{M}^S$, we advocate to find $\boldsymbol{x}^{adv}$ located at flat local region. That means we encourage that not only $\boldsymbol{x}^{adv}$ itself has low loss value, but also the points in the vicinity of $\boldsymbol{x}^{adv}$ have similarly low loss values.

To this end, we propose to minimize the maximal loss value within a local neighborhood region around the adversarial example $\boldsymbol{x}^{adv}$. The maximal loss is implemented by perturbing $\boldsymbol{x}^{adv}$ to maximize the attack loss, named *Reverse Adversarial Perturbation* (RAP). By inserting the RAP into the formulation (1), we aim to solve the following problem,

$$\min_{\boldsymbol{x}^{adv} \in \mathcal{B}_\epsilon(\boldsymbol{x})} \mathcal{L}(\mathcal{M}^s(\mathcal{G}(\boldsymbol{x}^{adv} + \boldsymbol{n}^{rap}); \boldsymbol{\theta}), y_t), \tag{2}$$

where

$$\boldsymbol{n}^{rap} = \operatorname*{arg\,max}_{\|\boldsymbol{n}^{rap}\|_\infty \le \epsilon_n} \mathcal{L}(\mathcal{M}^s(\boldsymbol{x}^{adv} + \boldsymbol{n}^{rap}; \boldsymbol{\theta}), y_t), \tag{3}$$

with $\boldsymbol{n}^{rap}$ indicating the RAP, and $\epsilon_n$ defining its search region. The above formulations Equation (2) and Equation (3) correspond to the targeted attack, and the corresponding untargeted formulations can be easily obtained by replacing the loss function $\mathcal{L}$ and $y_t$ by $-\mathcal{L}$ and $y$, respectively.

It is a min-max bi-level optimization problem [25], and can be solved by iteratively optimizing the inner maximization and the outer minimization problem. Specifically, in each iteration, given $\boldsymbol{x}^{adv}$, the inner maximization *w.r.t.* $\boldsymbol{n}^{rap}$ is solved by the projected gradient ascent algorithm:

$$\boldsymbol{n}^{rap} \leftarrow \boldsymbol{n}^{rap} + \alpha_n \cdot \text{sign}(\nabla_{\boldsymbol{n}^{rap}} \mathcal{L}(\mathcal{M}^s(\boldsymbol{x}^{adv} + \boldsymbol{n}^{rap}; \boldsymbol{\theta}), y_t)). \tag{4}$$

The above update is conducted by $T$ steps, and $\alpha_n = \frac{\epsilon_n}{T}$. Then, given $\boldsymbol{n}^{rap}$, the outer minimization *w.r.t.* $\boldsymbol{x}^{adv}$ can be solved by any off-the-shelf algorithm that is developed for solving Equation (1). For example, it can be undated by one step projected gradient descent, as follows:

$$\boldsymbol{x}^{adv} \leftarrow \text{Clip}_{\mathcal{B}_\epsilon(\boldsymbol{x})}\left[\boldsymbol{x}^{adv} - \alpha \cdot \text{sign}(\nabla_{\boldsymbol{x}^{adv}} \mathcal{L}(\mathcal{M}^s(\mathcal{G}(\boldsymbol{x}^{adv} + \boldsymbol{n}^{rap}); \boldsymbol{\theta}), y_t))\right], \tag{5}$$

with $\text{Clip}_{\mathcal{B}_\epsilon(\boldsymbol{x})}(\boldsymbol{a})$ clipping $\boldsymbol{a}$ into the neighborhood region $\mathcal{B}_\epsilon(\boldsymbol{x})$. The overall optimization procedure is summarized in Algorithm 1. Moreover, since the optimization *w.r.t.* $\boldsymbol{x}^{adv}$ can be implemented by any off-the-shelf algorithm for solving Problem Equation (1), one notable advantage of the proposed RAP is that it can be naturally combined with any one of them, such as the input transformation methods [6, 24, 41, 48].

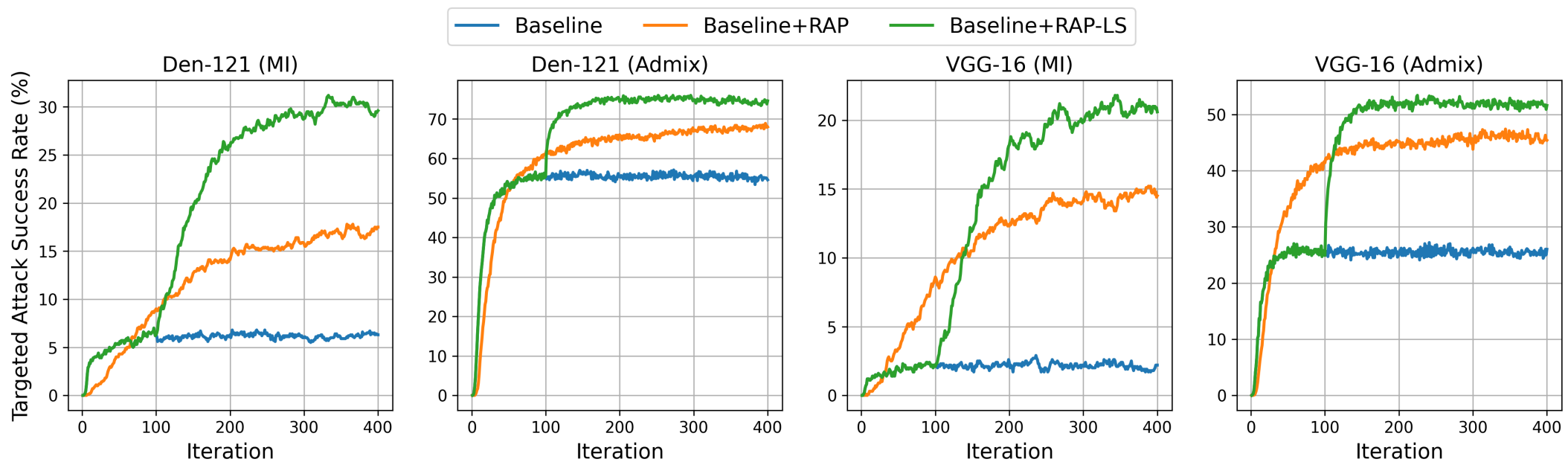

Figure 2: Targeted attack success rate (%) on Dense-121 and VGG-16. We take the Res-50 as the surrogate model and take MI and Admix as baseline methods.

**Algorithm 1** Reverse Adversarial Perturbation (RAP) Algorithm

**Input:** Surrogate model $\mathcal{M}^s$, benign data $(\boldsymbol{x}, y)$, target label $y_t$, loss function $\mathcal{L}$, transformation $\mathcal{G}$, the global iteration number $K$, the late-start iteration number $K_{LS}$ of RAP, as well as hyper-parameters in optimization (specified in later experiments) **Output:** the adversarial example $\boldsymbol{x}^{adv}$

**Initialize** $\boldsymbol{x}^{adv} \leftarrow \boldsymbol{x}$
**for** $k = 1, \ldots, K$ **do**
  **if** $k \geq K_{LS}$ **then**
    **Initialize** $\boldsymbol{n}^{rap} \leftarrow \mathbf{0}$
    **for** $t = 1, \ldots, T$ **do**
      Update $\boldsymbol{n}^{rap}$ using Equation (4)
  Update $\boldsymbol{x}^{adv}$ using Equation (5)

**A Late-Start (LS) Variant of RAP.** In our preliminary experiments, we find that RAP requires more iterations to converge and the performance is slightly lower during the initial iterations, compared to its baseline attack methods. As shown in Figure 2, we combine MI [5] and Admix [41] with RAP, and adopt ResNet-50 as the surrogate model. We take the evaluation on 1000 images from ImageNet (see Sec.4.1). It is observed that the method with RAP (see the orange curves) quickly surpasses its baseline method (see the blue curves) and finally achieves much higher success rate with more iterations, which verify the effect of RAP on enhancing the adversarial transferability. However, it is also observed that the performance of RAP is slightly lower than its baseline method in the early stage. The possible reason is that the early-stage attack is of very weak attack performance to the surrogate model. In this case, it may be waste to pursue better transferable attacks by solving the min-max problem. A better strategy may be only solving the minimization problem Equation (1) in the early stage to quickly achieve the region of relatively high adversarial attack performance, then starting RAP to further enhance the attack performance and transferability simultaneously. This strategy is denoted as RAP with late-start (RAP-LS), whose effect is preliminarily supported by the results shown in Figure 2 (see the green curve) and will be evaluated extensively in later experiments.

### 3.3 A Closer Look at RAP

To verify whether RAP can help us find a $\boldsymbol{x}^{adv}$ located at the local flat region or not, we use ResNet-50 as surrogate model and conduct the untargeted attacks. We visualize the loss landscape around $\boldsymbol{x}^{adv}$ on $\mathcal{M}^S$ by plotting the loss variations when we move $\boldsymbol{x}^{adv}$ along a random direction with different magnitudes $a$. The details of the calculation are provided in *Appendix*. Figure 3 plots the visualizations. We take I-FGSM [21] (denoted as I), MI [5], DI [48], and MI-TI-DI (MTDI) as baselines attacks and combined them with RAP. We can see that comparing to the baselines, RAP could help find $\boldsymbol{x}^{adv}$ located at the flat region.

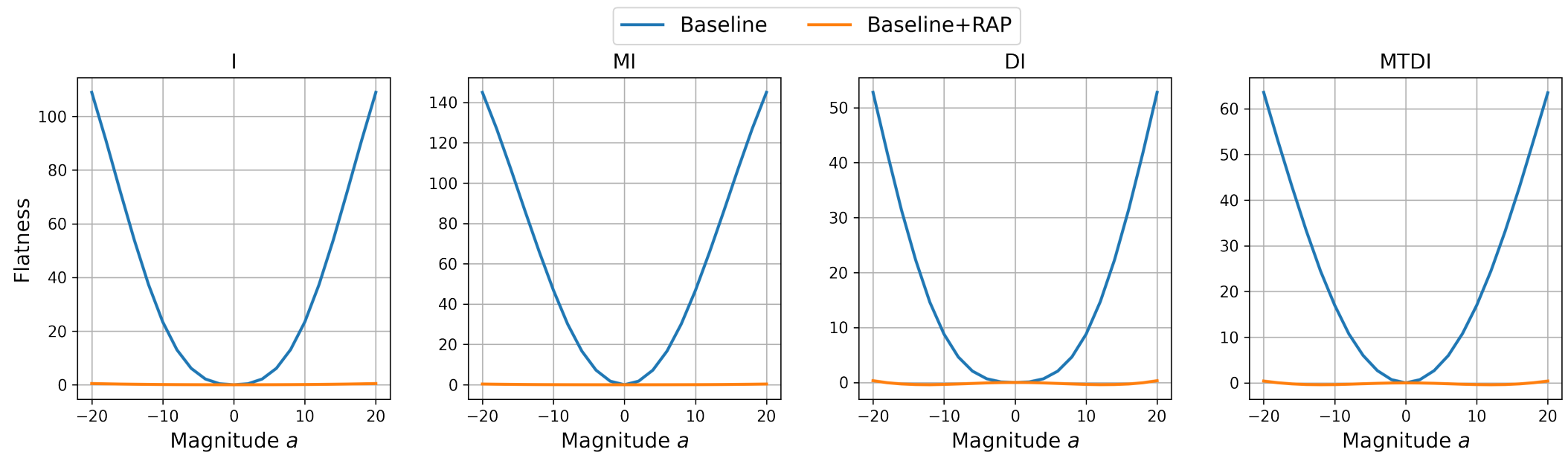

Figure 3: The flatness visualization of untargeted adversarial examples on $\mathcal{M}^S$. The implementation details are shown in Section B of *Appendix*.

# 4 Experiments

## 4.1 Experimental Settings

**Dataset and Evaluated Models.** We conduct the evaluation on the ImageNet-compatible dataset [1] comprised of 1,000 images. For the surrogate models, we consider the four widely used network architectures: Inception-v3 (Inc-v3) [36], ResNet-50 (Res-50) [15], DenseNet-121 (Dense-121) [16], and VGG-16bn (VGG-16) [34]. For target models, apart from the above models, we also utilize more diverse architectures: Inception-ResNet-v2 (Inc-Res-v2) [37], NASNet-Large (NASNet-L) [51], and ViT-Base/16 (ViT-B/16) [8]. For defense models, we adopt the two widely used ensemble adversarial training (AT) models: adv-Inc-v3 (Inc-v3$_{adv}$) and ens-adv-Inc-Res-v2 (IncRes-v2$_{ens}$) [38]. Besides, we also test multi-step AT model [33], imput transformation defense [46], feature denoising [47], and purification defense (NRP) [29].

**Compared Methods.** We adopt I-FGSM [21] (denoted as I), MI [5], TI [6], DI [48], SI [24], Admix [41], VT [40], EMI [42], ILA [17], LinBP [14], Ghost Net [22], and the generative targeted attack method TTP [30]. We also consider the combination of baseline methods, including MI-TI-DI (MTDI), MI-TI-DI-SI (MTDSI), and MI-TI-DI-Admix (MTDAI). Besides, Expectation of Transformation (EOT) method [1] is also a comparable baseline method. We also conduct the comparison of RAP and EOT.

**Implementation Details.** For untargeted attack, we adopt the Cross Entropy (CE) loss. For targeted attack, apart from CE, we also experiment with the logit loss, where Zhao et al. [49] shows it behaves better for targeted attack. The adversarial perturbation $\epsilon$ is restricted by $\ell_\infty = 16/255$. The step size $\alpha$ is set as $2/255$ and number of iteration $K$ is set as 400 for all attacks. In the following, we mainly show the results at $K = 400$ and the results at different value of $K$ are shown in *Appendix*. For RAP, we set $K_{LS}$ as 100 and $\alpha_n$ as $2/255$. We set $\epsilon_n$ as $12/255$ for I and TI in untargeted attack and $16/255$ for other attacks in all other settings. The computational cost is shown in Section B of *Appendix*.

**Extra Experiments in Appendix.** Due to the space limitation, we put extra experiment results in *Appendix*. The comparisons of RAP and EOT, VT, EMI, and Ghost Net methods are shown in Section C in *Appendix*. The evaluation of RAP on stronger defense model, multi-step AT models, NRP, and feature denoising, is shown in Section D. The evaluation of ensemble-model attacks on these diverse network architectures is given in Section E.2.

## 4.2 The Evaluation of Untargeted Attacks

**Baseline Methods.** We first evaluate the performance of RAP and RAP-LS with different baseline attacks, including I, MI, DI, TI, SI, and Admix. The results are shown in Table 1. For instance, the 'MI/ +RAP/ +RAP-LS' denotes the methods of baseline MI, MI+RAP, and MI+RAP-LS, respectively. RAP achieves the significant improvements for all methods on each target model. For average attack success rate of all target models, RAP outperforms the I and MI by 9.6% and 16.3%, respectively.

[1]Publicly available from https://github.com/cleverhans-lab/cleverhans/tree/master/cleverhans_v3.1.0/examples/nips17_adversarial_competition/dataset

Table 1: The **untargeted attack success rate (%) of baseline attacks with RAP**. The results with $CE$ loss are reported. The best results are bold and the second best results are underlined.

| Attack | ResNet-50 ⟹ Dense-121 | ResNet-50 ⟹ VGG-16 | ResNet-50 ⟹ Inc-v3 | DenseNet-121⟹ Res-50 | DenseNet-121⟹ VGG-16 | DenseNet-121⟹ Inc-v3 |
|---|---|---|---|---|---|---|
| I / +RAP / +RAP-LS | 79.2 / 91.5 / **91.9** | 78.0 / 91.1 / **92.9** | 34.6 / 57.0 / **57.2** | 87.4 / 94.2 / **94.3** | 85.1 / 91.7 / **92.8** | 46.5 / 60.2 / **61.1** |
| MI / +RAP / +RAP-LS | 85.8 / 95.0 / **96.1** | 82.4 / 93.9 / **94.5** | 50.3 / 75.9 / **77.4** | 90.3 / 97.6 / **97.9** | 87.5 / 96.0 / **97.6** | 59.3 / 80.4 / **82.8** |
| TI / +RAP / +RAP-LS | 82.0 / 94.1 / **95.1** | 81.0 / 93.1 / **93.3** | 45.5 / 66.1 / **67.0** | 89.6 / 94.2 / **94.8** | 87.0 / 92.1 / **93.3** | 54.2 / 66.7 / **70.0** |
| DI / +RAP / +RAP-LS | 99.0 / 99.6 / **99.7** | 99.0 / 99.6 / **99.7** | 57.7 / 82.9 / **85.0** | 98.2 / 99.6 / **99.7** | 98.1 / **99.4** / **99.4** | 67.6 / 86.6 / **86.9** |
| SI / +RAP / +RAP-LS | 94.9 / 98.9 / **99.7** | 88.6 / 95.7 / **97.2** | 65.9 / 79.7 / **84.4** | 95.1 / 96.9 / **98.8** | 91.9 / 95.0 / **97.5** | 71.6 / 83.2 / **87.4** |
| Admix / +RAP / +RAP-LS | 97.9 / 99.6 / **99.9** | 95.8 / 97.7 / **99.0** | 77.7 / 87.4 / **92.6** | 97.0 / 99.0 / **99.2** | 95.6 / 97.7 / **98.6** | 82.0 / 89.8 / **93.8** |
| **Attack** | **VGG-16 ⟹ Res-50** | **VGG-16 ⟹ Dense-121** | **VGG-16 ⟹ Inc-v3** | **Inc-v3⟹ Res-50** | **Inc-v3⟹ Dense-121** | **Inc-v3⟹ VGG-16** |
| I / +RAP / +RAP-LS | 53.7 / 53.0 / **54.2** | 49.1 / 50.6 / **51.4** | 22.0 / 24.7 / **24.9** | 51.5 / **62.1** / 62.0 | 48.7 / **60.8** / 60.0 | 55.1 / 65.9 / **68.0** |
| MI / +RAP / +RAP-LS | 62.5 / 76.2 / **76.4** | 60.5 / 73.0 / **73.9** | 30.0 / **42.7** / 42.2 | 62.0 / **85.8** / 84.8 | 56.7 / **84.6** / **84.6** | 63.1 / **84.9** / 84.6 |
| TI / +RAP / +RAP-LS | 62.8 / 64.8 / **65.8** | 55.9 / **63.7** / 62.1 | 29.1 / 36.2 / **37.1** | 49.3 / **63.4** / 61.6 | 49.4 / 63.4 / **63.8** | 58.1 / 68.6 / **69.5** |
| DI / +RAP / +RAP-LS | 72.2 / 86.0 / **88.8** | 68.8 / 85.0 / **87.4** | 29.9 / 46.6 / **51.6** | 68.4 / 81.7 / **81.8** | 71.9 / **85.0** / 84.0 | 76.1 / 85.2 / **86.4** |
| SI / +RAP / +RAP-LS | 80.0 / 92.7 / **94.7** | 82.1 / 94.8 / **95.7** | 45.8 / 74.0 / **74.7** | 66.2 / 69.8 / **72.8** | 65.9 / 74.9 / **77.2** | 66.0 / 69.2 / **73.0** |
| Admix / +RAP / +RAP-LS | 87.3 / 94.6 / **96.8** | 88.2 / 96.4 / **97.2** | 55.5 / 77.6 / **80.8** | 75.9 / 80.2 / **84.9** | 78.5 / 83.7 / **87.4** | 74.5 / 77.2 / **83.5** |

Table 2: The **untargeted attack success rate (%) of combinational methods with RAP**. The results with $CE$ loss are reported. The best results are bold and the second best results are underlined.

| Attack | ResNet-50 ⟹ Dense-121 | ResNet-50 ⟹ VGG-16 | ResNet-50 ⟹ Inc-v3 | DenseNet-121⟹ Res-50 | DenseNet-121⟹ VGG-16 | DenseNet-121⟹ Inc-v3 |
|---|---|---|---|---|---|---|
| MTDI / +RAP / +RAP-LS | 99.8 / **100** / **100** | 99.8 / **100** / 99.9 | 85.7 / 96.0 / **96.9** | 99.4 / 99.8 / **100** | 99.2 / 99.5 / **100** | 89.1 / **97.1** / **97.1** |
| MTDSI / +RAP / +RAP-LS | **100** / **100** / **100** | 99.7 / **99.9** / 99.8 | 97.0 / **99.1** / **99.1** | 99.8 / **99.9** / **99.9** | 99.2 / 99.3 / **99.7** | 95.1 / 98.3 / **98.4** |
| MTDAI / +RAP / +RAP-LS | **100** / **100** / **100** | 99.8 / **99.9** / **99.9** | 98.3 / 99.2 / **99.8** | 99.8 / 99.8 / **99.9** | 99.4 / 99.6 / **99.8** | 97.9 / 98.8 / **98.9** |
| **Attack** | **VGG-16 ⟹ Res-50** | **VGG-16 ⟹ Dense-121** | **VGG-16 ⟹ Inc-v3** | **Inc-v3⟹ Res-50** | **Inc-v3⟹ Dense-121** | **Inc-v3⟹ VGG-16** |
| MTDI / +RAP / +RAP-LS | 90.0 / 97.2 / **97.7** | 88.8 / 97.0 / **97.3** | 56.8 / **82.6** / 81.4 | 82.9 / **91.8** / 90.6 | 85.7 / **94.2** / 93.3 | 85.1 / **92.7** / 91.0 |
| MTDSI / +RAP / +RAP-LS | 97.6 / 98.8 / **99.4** | 98.1 / 99.2 / **99.4** | 85.0 / 94.1 / **95.2** | 89.0 / 91.2 / **92.3** | 92.0 / 95.2 / **95.6** | 87.6 / 90.3 / **92.2** |
| MTDAI / +RAP / +RAP-LS | 97.8 / 99.2 / **99.6** | 98.9 / 99.5 / **99.6** | 89.3 / 95.0 / **95.5** | 91.5 / 94.1 / **94.7** | 95.4 / 96.2 / **97.6** | 91.4 / 93.2 / **94.1** |

For TI, DI, SI, and Admix, RAP gets the improvements by 10.2%, 10.9%, 9.3%, and 6.3%. With late-start, RAP-LS further enhance the transfer attack performance for almost all methods.

**Combinational Methods.** Prior works demonstrate the combination of baseline methods could largely boost the adversarial transferability [41, 49]. We also investigate of behavior of RAP when incorporated with the combinational attacks. The results are shown in Table 2. As shown in the table, there exist the clear improvements of the combinational attacks over all baseline attacks shown in Table 1. In addition, our RAP-LS further boosts the average attack success rate of the three combinational attacks by 6.9%, 2.6%, and 1.7% respectively. Combined with the three combinational attacks, RAP-LS achieves 95.4%, 97.6%, and 98.3% average attack success rate, respectively. These results demonstrate RAP can significantly enhance the transferability.

### 4.3 The Evaluation of Targeted Attacks

We then evaluate the targeted attack performance of the different methods with RAP. The results with logit loss are presented and the results with CE loss are shown in *Appendix*.

**Baseline Methods.** The results of RAP with baseline attacks are shown in Table 3. From the results, RAP is also very effective in enhancing the transferability in targeted attacks. Taking ResNet-50 and DenseNet-121 as surrogate models for example, the average performance improvements induced by RAP are 5.0% (I), 8.1% (MI), 4.6% (TI), 10.4% (DI), 18.5% (SI), and 15.1% (Admix), respectively. Comparing to the ResNet-50 and DenseNet-121, the baseline attacks generally achieve lower transferability when using the VGG-16 or Inception-v3 as the surrogate models, which has also been verified in existing works [48, 49]. However, for Inception-v3 and VGG-16 as the surrogate models, RAP also consistently boosts the transferability under all cases. With late-start, RAP-LS could further improve the transferability of RAP for most attacks. The average attack success rate under all attack cases of RAP-LS is 2.6% higher than that of RAP.

**Combinational Methods.** As did in the untargeted attacks, we also evaluate the performance of combinational methods. The results are shown in Table 4. Similar to the findings in untargeted attacks, the combinational methods obtain significantly improvements over baseline methods. The RAP-LS outperforms all combinational methods by a significantly margin. For example, taking the average attack success rate of all target models as evaluation metric, RAP-LS obtains 14.2%, 11.8%, 9.3% improvements over the MTDI, MTDSI and MTDAI, respectively.

Table 3: The **targeted attack success rate** (%) **of baseline methods with RAP**. The results with logit loss are reported. The best results are bold and the second best results are underlined.

| Attack | ResNet-50 ⟹ Dense-121 | ResNet-50 ⟹ VGG-16 | ResNet-50 ⟹ Inc-v3 | DenseNet-121⟹ Res-50 | DenseNet-121⟹ VGG-16 | DenseNet-121⟹ Inc-v3 |
|---|---|---|---|---|---|---|
| I / +RAP / +RAP-LS | 4.5 / 9.5 / **14.3** | 2.4 / 9.8 / **11.8** | 0.1 / 0.1 / **0.7** | 5.0 / 12.8 / **17.9** | 2.9 / 10.1 / **15.9** | 0.0 / 0.8 / **1.2** |
| MI / +RAP / +RAP-LS | 6.3 / 17.5 / **29.6** | 2.2 / 14.5 / **20.6** | 0.1 / 1.1 / **2.4** | 4.6 / 16.2 / **26.5** | 3.1 / 13.4 / **23.2** | 0.3 / 2.0 / **3.4** |
| TI / +RAP / +RAP-LS | 7.2 / 11.0 / **17.3** | 4.0 / 12.9 / **15.3** | 0.1 / 0.8 / **1.2** | 8.4 / 13.5 / **20.8** | 5.2 / 12.4 / **16.4** | 0.2 / 2.1 / **3.0** |
| DI / +RAP / +RAP-LS | 62.6 / 64.9 / **73.9** | 57.2 / 63.4 / **69.3** | 1.5 / 7.9 / **10.1** | 30.2 / 52.6 / **60.4** | 32.1 / 49.5 / **58.9** | 1.4 / 8.8 / **10.0** |
| SI / +RAP / +RAP-LS | 30.0 / 53.2 / **61.1** | 9.5 / 32.8 / **36.0** | 1.8 / 9.3 / **10.5** | 14.2 / 41.5 / **43.4** | 8.4 / 31.0 / **35.2** | 1.6 / 8.5 / **10.4** |
| Admix / +RAP / +RAP-LS | 54.6 / 68.0 / **74.6** | 26.0 / 45.4 / **51.6** | 5.8 / 17.1 / **19.6** | 29.3 / 53.0 / **58.2** | 21.5 / 42.7 / **48.2** | 5.0 / 17.1 / **17.6** |
| **Attack** | **VGG-16 ⟹ Res-50** | **VGG-16 ⟹ Dense-121** | **VGG-16 ⟹ Inc-v3** | **Inc-v3⟹ Res-50** | **Inc-v3⟹ Dense-121** | **Inc-v3⟹ VGG-16** |
| I / +RAP / +RAP-LS | 0.1 / 0.7 / **1.4** | 0.2 / 1.4 / **1.7** | 0.0 / 0.1 / **0.2** | 0.2 / **0.9** / 0.5 | 0.2 / **0.6** / 0.3 | 0.1 / **0.5** / **0.5** |
| MI / +RAP / +RAP-LS | 0.5 / 1.3 / **1.9** | 0.5 / 2.3 / **3.0** | 0.0 / 0.0 / **0.3** | 0.2 / **1.7** / 1.5 | 0.1 / **1.6** / 1.5 | 0.2 / **1.3** / 1.0 |
| TI / +RAP / +RAP-LS | 0.7 / 1.2 / **3.2** | 0.8 / 1.7 / **2.9** | 0.0 / 0.1 / **0.4** | 0.2 / 0.5 / **0.7** | 0.1 / **0.7** / 0.6 | 0.2 / **0.8** / 0.6 |
| DI / +RAP / +RAP-LS | 2.8 / 7.3 / **9.7** | 3.8 / 8.4 / **12.7** | 0.0 / 0.4 / **1.1** | 1.6 / 4.6 / **6.4** | 2.8 / 5.8 / **7.5** | 2.6 / 6.3 / **8.1** |
| SI / +RAP / +RAP-LS | 3.3 / **9.8** / **9.8** | 7.2 / 16.8 / **17.8** | 0.2 / 1.7 / **1.8** | 0.6 / **2.9** / 2.5 | 0.9 / 2.7 / **3.2** | 0.5 / 1.5 / **2.3** |
| Admix / +RAP / +RAP-LS | 5.6 / 11.1 / **11.9** | 13.0 / 20.2 / **23.6** | 0.7 / 2.4 / **2.8** | 1.5 / 4.9 / **5.2** | 2.0 / 6.9 / **7.5** | 1.3 / 3.3 / **4.4** |

Table 4: The **targeted attack success rate** (%) **of combinational methods with RAP**. The results with logit loss are reported. The best results are bold and the second best results are underlined.

| Attack | ResNet-50 ⟹ Dense-121 | ResNet-50 ⟹ VGG-16 | ResNet-50 ⟹ Inc-v3 | DenseNet-121⟹ Res-50 | DenseNet-121⟹ VGG-16 | DenseNet-121⟹ Inc-v3 |
|---|---|---|---|---|---|---|
| MTDI / +RAP / +RAP-LS | 74.9 / 78.2 / **88.5** | 62.8 / 72.9 / **81.5** | 10.9 / 28.3 / **33.2** | 44.9 / 64.3 / **74.5** | 38.5 / 55.0 / **65.5** | 7.7 / 23.0 / **26.5** |
| MTDSI / +RAP / +RAP-LS | 86.3 / 88.4 / **93.3** | 70.1 / 77.7 / **84.7** | 38.1 / 51.8 / **58.0** | 55.0 / 71.2 / **75.8** | 42.0 / 58.4 / **62.3** | 19.8 / 39.0 / **39.2** |
| MTDAI / +RAP / +RAP-LS | 91.4 / 89.4 / **93.6** | 79.9 / 79.0 / **86.3** | 50.8 / 57.1 / **64.1** | 69.1 / 74.2 / **82.1** | 54.7 / 63.1 / **69.3** | 32.0 / 43.5 / **49.3** |
| **Attack** | **VGG-16 ⟹ Res-50** | **VGG-16 ⟹ Dense-121** | **VGG-16 ⟹ Inc-v3** | **Inc-v3⟹ Res-50** | **Inc-v3⟹ Dense-121** | **Inc-v3⟹ VGG-16** |
| MTDI / +RAP / +RAP-LS | 11.8 / 16.7 / **22.9** | 13.7 / 19.4 / **27.4** | 0.7 / 3.4 / **4.6** | 1.8 / **8.3** / 7.5 | 4.1 / **14.8** / 13.4 | 2.9 / 8.0 / **9.8** |
| MTDSI / +RAP / +RAP-LS | 31.0 / 35.3 / **38.7** | 41.7 / 44.4 / **49.6** | 9.6 / **15.2** / 13.7 | 5.6 / **11.9** / 10.7 | 10.4 / **21.2** / 20.9 | 4.2 / **8.9** / 8.6 |
| MTDAI / +RAP / +RAP-LS | 36.2 / 39.0 / **43.1** | 48.0 / 45.1 / **55.2** | 11.6 / 17.1 / **17.6** | 9.6 / 13.6 / **16.7** | 17.9 / 27.5 / **31.6** | 8.4 / 12.0 / **12.1** |

### 4.4 The Comparison with Other Types of Attacks

Apart from the baseline and the combinational methods, we also experiment with more diverse attack methods, including the model-specific attack LinBP [14], the feature-based attack ILA [17], and the generative targeted attack TTP [30]. The LinBP depends on the skip connection and the authors only provide the source code about ResNet-50. We use their released code and thus conduct experiments with ResNet-50 as $\mathcal{M}^S$. The results of LibBP and ILA are shown in Table 5, where we also implement the variants of LinBP following Guo et al. [14], inlcuding LinBP-ILA, LinBP-ILA-SGM, LinBP-MI-DI, and LinBP-MI-DI-SGM. We observe that our MI-DI-RAP significantly outperforms the LinBP and ILA, especially for the targeted attacks. Compared with the second-best method (*i.e.*, LinBP-MI-DI-SGM), we obtain a large improvement by 33.5% on average ASR of targeted attacks.

Table 5: The comparison with ILA and LinBP. We use ResNet-50 as $\mathcal{M}^S$. The best results are bold.

| Attack | Untarged Dense-121 | Untarged VGG-16 | Untarged Inc-v3 | Targeted Dense-121 | Targeted VGG-16 | Targeted Inc-v3 |
|---|---|---|---|---|---|---|
| ILA | 95.0 | 94.2 | 77.7 | 2.8 | 1.5 | 0.5 |
| LinBP-ILA | 99.5 | 99.2 | 89.8 | 9.4 | 4.9 | 2.0 |
| LinBP-ILA-SGM | 99.7 | 99.3 | 91.1 | 13.3 | 7.2 | 2.8 |
| LinBP-MI-DI | 99.5 | 99.2 | 89.3 | 26.1 | 16.5 | 3.2 |
| LinBP-MI-DI-SGM | 99.8 | 99.3 | 90.2 | 32.6 | 22.1 | 4.6 |
| MI-DI+RAP | **99.9** | **100** | **93.7** | **75.1** | **69.7** | **13.9** |

TTP [30] is the state-of-the-art generative method to conduct targeted attack. To compare with it, we adopt the generators based on ResNet-50 provided by the authors. Since TTP needs to train the perturbation generator for each targeted class, we follow their "10-Targets (all-source)" setting, as did in Zhao et al. [49]. The results are shown in Table 6, where our MTDSI+RAP-LS behaves best and outperforms TTP and MTDI by large margins of 14.9% and 25.7%, respectively.

Table 6: The comparison with TTP on targeted attack. The best results are bold.

| Attack | Dense-121 | VGG-16 | Inc-v3 |
|---|---|---|---|
| TTP | 79.6 | 78.6 | 40.3 |
| MTDI | 78.6 | 74.6 | 12.7 |
| MTDI+RAP-LS | 90.8 | 87.2 | 35.4 |
| MTDSI | 93.2 | 80.0 | 41.3 |
| MTDSI+RAP-LS | **95.7** | **88.1** | **59.3** |

### 4.5 The Evaluation on Diverse Network Architectures and Defense Models

To further demonstrate the efficacy of RAP, we evaluate our method on more diverse network architectures, including Inception-ResNet-v2, NASNet-Large and ViT-Base/16. We adopt ResNet-50 as the surrogate model and the results are shown in Table 7, *col 2-7*. As shown in the table, the

Table 7: The evaluation on diverse network architectures and defense models.

| Attack | Untarged | | | Targeted | | | Untarged | | Targeted | |
|---|---|---|---|---|---|---|---|---|---|---|
| | IncRes-v2 | NASNet-L | ViT-B/16 | IncRes-v2 | NASNet-L | ViT-B/16 | Inc-v3$_{adv}$ | IncRes-v2$_{ens}$ | Inc-v3$_{adv}$ | IncRes-v2$_{ens}$ |
| MTDI | 83.4 | 89.0 | 27.9 | 14.8 | 32.1 | 0.4 | 68.1 | 50.9 | 0.8 | 0.0 |
| MTDI+RAP-LS | **95.6** | **97.5** | **42.7** | **43.0** | **62.5** | **1.7** | **86.5** | **72.3** | **9.7** | **4.1** |
| MTDSI | 95.7 | 98.0 | 43.0 | 45.5 | 67.9 | 2.6 | 90.0 | 79.6 | 12.7 | 6.7 |
| MTDSI+RAP-LS | **98.6** | **99.7** | **57.4** | **64.0** | **80.4** | **5.3** | **96.5** | **91.5** | **31.0** | **22.0** |
| MTDAI | 97.3 | 98.8 | 45.5 | 58.4 | 75.3 | 3.3 | 92.1 | 82.7 | 17.2 | 12.2 |
| MTDAI+RAP-LS | **99.2** | **99.8** | **60.2** | **70.4** | **82.6** | **7.4** | **96.7** | **91.6** | **34.4** | **26.0** |

proposed RAP-LS achieves significant improvements for all three combinational methods on all target models, and MTDAI+RAP-LS achieves the best performance for diverse models. For MTDAI, the average performance improvements induced by RAP-LS is 5.9% and 7.8% for untargeted and targeted attacks, respectively. Since ViT is based on the transformer architecture that totally being different from convolution models, the transfer attacks based on Resnet-50 behave relatively poor on it, especially on targeted attacks. Yet our RAP-LS still obtains consistent improvements for all compared methods. We also consider the ensemble-model attack on these diverse network architectures and the results are given in *Appendix.*

Furthermore, we evaluate RAP on attacking defense models. We choose ensemble adversarial training (AT) model, adv-Inc-v3 and ens-adv-Inc-Res-v2 [38], multi-step AT model [33], imput transformation defense [46], feature denoising [47], and purification defense (NRP) [29]. We only demonstrate the results of ensemble AT in main submission. The evaluations of other defense models are shown in *Appendix*. Following prior works [41, 48], we adopt the ensemble-model attack by averaging the logits of different surrogate models, including ResNet-50, ResNet-101, Inception-v3, and Inception-ResNet-v2. The transfer attack success rate on defense models are shown in Table 7, *col 8-11*. We can observe that our RAP-LS further boosts transferability of the baseline methods on both targeted and untargeted attacks. For untargeted attacks, RAP-LS achieves average performance improvements of $9.8\%$ and $14.1\%$ on Inc-v3$_{adv}$ and IncRes-v2$_{ens}$, respectively. For targeted attacks, the average performance improvements of RAP-LS are $14.8\%$ and $11.1\%$, respectively.

### 4.6 Ablation Study

We conduct ablation study on the hyper-parameters of the proposed RAP, including the size of neighborhoods $\epsilon_n$, the iteration number of inner optimization $T$ and late-start $K_{LS}$. We adopt targeted attacks with ResNet-50 as the surrogate model.

We first evaluate the effect of $\epsilon_n$ and $T$. We consider different values of $\epsilon_n$, including $2/255$, $4/255$, $8/255$, $12/255$, $16/255$, and $20/255$. In Figure 4 (a), we plot the tendency curves of the targeted attack success rate under different values of $\epsilon_n$ and $T$. Note that in Sec. 3.2, we set $\alpha_n = \epsilon_n/T$. Thus for a fixed $\epsilon_n$, larger $T$ indicating lower stepsize $\alpha_n$. The minimum stepsize of $\alpha_n$ is set to $2/255$. We have the following observation from the plot: for a fixed $\epsilon_n$, the more iterations $T$, the better attack performance. Thus, we adopt a relatively smaller $\alpha_n = 2/255$ in our experiments. In Figure 4 (b-d), we further plot the results of different attack methods and target models *w.r.t.* $\epsilon_n$, where $\alpha_n = 2/255$. As shown in the plots, the larger $\epsilon_n$ generally improves the attack performance. For Inception-v3 and DenseNet-121, the improvements become mild for even larger $\epsilon_n$. Overall, the value of 12 or 16 could lead to satisfactory result under most cases.

Then we conduct the ablation study of $K_{LS}$. In Figure 5, we report the targeted attack success rate of I, MI, DI, and MI-TI-DI combined with RAP-LS with $K_{LS} = 0, 25, 50, 100, 150, 200$. Note that the RAP-LS with $K_{LS} = 0$ reduces to RAP. As shown in the plots, the proposed late-start strategy can further boost attack performance of RAP for most cases. In general, the performance improvements increase as $K_{LS}$ increases, and then become mild when $K_{LS}$ is larger than 100. The suitable value of $K_{LS}$ is relatively consistent among different methods and target models.

### 4.7 The Targeted Attack Against Google Cloud Vision API

Finally, we conduct the transfer attacks to attack a practical and widely used image recognition system, Google Cloud Vision API, and in the more challenging targeted attack scenario. MTDAI-RAP-LS behaves the best performance in above experiments, so we choose it to conduct the attack. We take the evaluation on randomly selected 500 images and use ResNet-50 as surrogate model. As the API

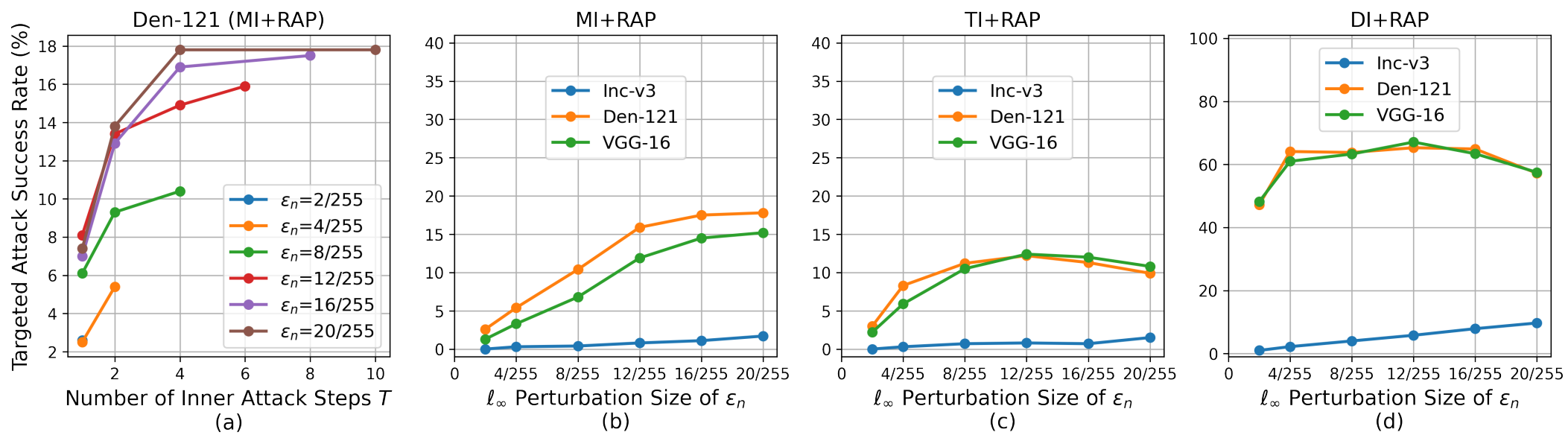

Figure 4: Targeted attack success rate (%) with various $T$ and $\epsilon_n$. Res-50 is set as surrogate model.

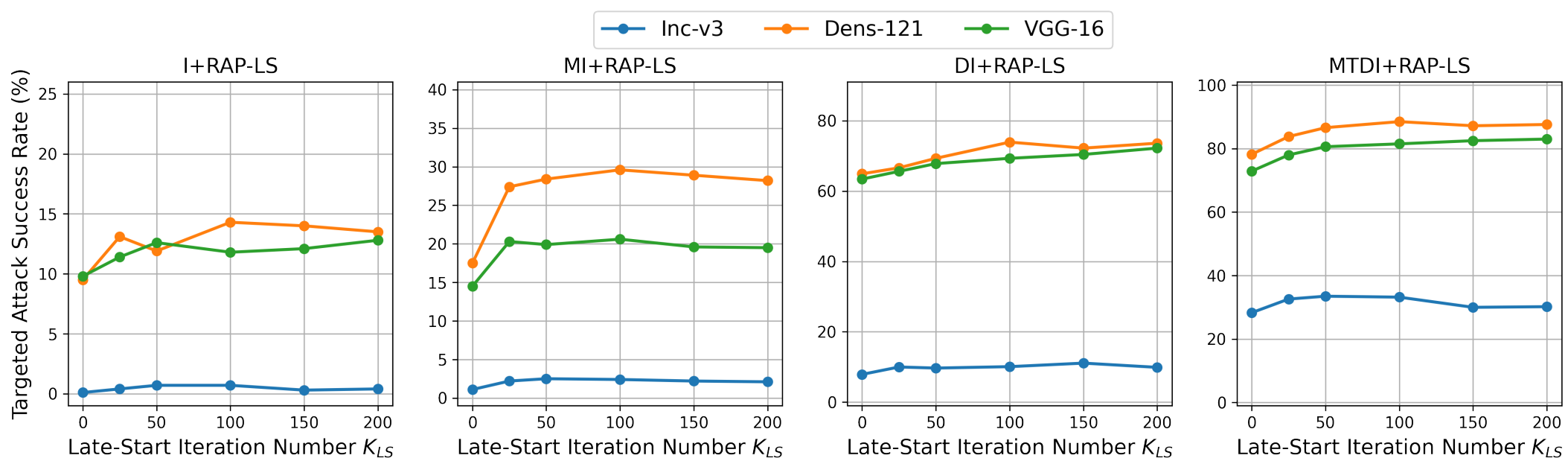

Figure 5: Targeted attack success rate (%) with various $K_{LS}$. Res-50 is set as surrogate model.

returns 10 predicted labels for each query, to evaluate the attacking performance, we test whether or not the target class appears in the returned predictions. Since the predicted label space of Google Cloud Vision API do not fully correspond to the 1000 ImageNet classes, we manually treat classes with similar semantics to be the same classes. In comparison, the baseline MTDAI successfully attacks 232 images against the Google API. Our RAP-LS achieves a large improvement, successfully attacking 342 images, leading to a 22.0% performance improvements. These demonstrates the high efficacy of our method to improve transferability on real-world system.

## 5 Conclusion

In this work, we study the transferability of adversarial examples that is significant for black-box attacks. The transferability of adversarial examples is generally influenced by the overfitting of surrogate models. To alleviate this, we propose to seeking adversarial examples that locate at flatter local regions. That is, instead of optimizing the pinpoint attack loss, we aim to obtain a consistently low loss at the neighbor regions of the adversarial examples. We formulate this as a min-max bi-level optimization problem, where the inner maximization aims to inject the worse-case perturbation for the adversarial examples. We conduct a rigorous experimental study, covering untargeted attack and targeted attack, standard and defense models, and a real-world Google Clould Vision API. The experimental results demonstrate that RAP can significantly boost the transferability of adversarial examples, which also demonstrates that transfer attacks have become serious threats. We need to consider how to effectively defense against them.

### Acknowledgments

We want to thank the anonymous reviewers for their valuable suggestions and comments. This work is supported by the National Natural Science Foundation of China under grant No.62076213, Shenzhen Science and Technology Program under grant No.RCYX20210609103057050, and the university development fund of the Chinese University of Hong Kong, Shenzhen under grant No.01001810, and sponsored by CCF-Tencent Open Fund.

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
