## A    Social Impact

Deep neural networks (DNNs) have been successfully applied in many safety-critical tasks, such as autonomous driving, face recognition and verification, *etc*. And adversarial samples have posed a serious threat to machine learning systems. For real-world applications, the DNN model as well as the training dataset, are often hidden from users. Therefore, the attackers need to generate the adversarial examples under black-box setting where they do not know any information of the target model. For black-box setting, the adversarial transferability matters since it can allow the attackers to attack target models by using adversarial examples generated on the surrogate models. This work can potentially contribute to understanding of transferability of adversarial examples. Besides, the better transferability of adversarial examples calls the machine learning and security communities into action to create stronger defenses and robust models against black-box attacks.

## B    Implementation Details

We conducted all experiments in an Nvidia-V100 GPU. And we run all experiments 3 times and average all results over 3 random seeds.

**Dataset**    The used two datasets are licensed under MIT. Imagenet is licensed under Custom (non-commercial).

**Implementation Details of Evaluated Models.**    For ResNet-50, DenseNet-121, VGG-16, Inception-v3, we adopt the pre-trained models provided by torchvision package. For Inception-ResNet-v2, NASNet-Large, ViT-Base/16, adv-Inc-v3, and ens-adv-Inc-Res-v2, we adopt the provided pre-trained models[2].

**Implementation Details of Baseline Attack Methods.**    We adopt the source code [3] provided by Zhao et al. [49] to implement I, MI, TI, and DI attacks. The decay factor for MI is set as $1.0$. The kernel size is set as $5$ for TI attack, following Gao et al. [11]. The transformation probability is set as $0.7$ for DI. For SI and Admix, we adopt the parameters suggested in Wang et al. [41]. The number of copies for SI is set as $5$. The number of randomly sample $m_2$ and $\eta$ of Admix are set as 3 and $0.2$ respectively. For implementation of ILA and LinBP, we utilize the source code [4] provided by Guo et al. [14]. For implementation of TTP, we use the pre-trained generator [5] based on ResNet-50 provided by [30].

**Computational Cost.**    Here, we analyze the computational cost of our method. In Algorithm 1 with global iteration number $K$, late-start iteration number $K_{LS}$ and inner iteration number $T$, our RAP-LS requires $K + (K - K_{LS}) * T$ forward and backward calculation. While the original attack algorithm requires $K$ forward and backward calculation. The extra computation cost of RAP-LS is $(K - K_{LS}) * T$ times forward and backward calculation.

The adversarial example generation process is conducted based on the offline surrogate models. Compared with this offline time cost, the attacking performance is much more important for black-box attacks. Besides, our late-start strategy could alleviate the time cost.

**Implementation Details of Visualization.**    We visualize the flatness of the loss landscape around $x^{adv}$ on surrogate model by plotting the loss change when moving $x^{adv}$ along a random direction with different magnitudes. Specially, we first sample $d$ from a Gaussian distribution and normalize it on a $\ell_2$ unit norm ball, $d \leftarrow \frac{d}{\|d\|_F}$. Then, we calculate the loss change (flatness) $f(a)$ with different magnitudes $a$,

$$f(a) = \mathcal{L}(\mathcal{M}^s(\mathcal{G}(x^{adv} + a \cdot d); \theta), y_t) - \mathcal{L}(\mathcal{M}^s(\mathcal{G}(x^{adv}); \theta), y_t). \tag{6}$$

Considering $d$ is randomly selected, we repeat the above calculation 20 times with different $d$ and take the averaged value to conduct the visualization.

---

[2]https://github.com/rwightman/pytorch-image-models
[3]https://github.com/ZhengyuZhao/Targeted-Tansfer
[4]https://github.com/qizhangli/linbp-attack
[5]https://github.com/Muzammal-Naseer/TTP

We also add the visualization results about targeted attacks.

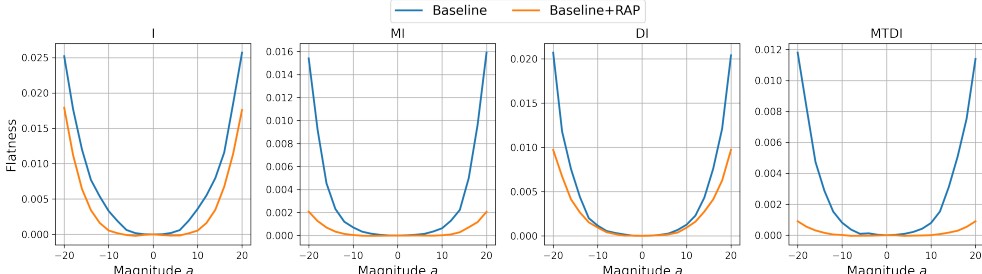

Figure 6: The flatness visualization of targeted adversarial examples.

## C Experimental Results about More baseline attacks

In this section, we show the comparison of our method RAP and EOT [1], VT [40], EMI [42], and Ghost Net [22] attack methods.

### C.1 Experimental Results about VT and EMI

In the below Table 8 and 9, we demonstrate the untargeted and targeted attack performance of VT, EMI, and our methods. We choose MI-TI-DI as the baseline method and follow experimental settings in Section 4.1. As shown in experimental results, our RAP-LS achieves better performance, especially for targeted attacks. Compared with VT, RAP-LS gets an increase of 6.7% for targeted attacks in terms of average success rate. This demonstrates the effectiveness of our methods.

Table 8: The untargeted attack success rate (%) of VT, EMI, and RAP-LS with the MI-TI-DI baseline.

| Attack | ResNet-50 $\implies$ | | | DenseNet-121$\implies$ | | |
| --- | --- | --- | --- | --- | --- | --- |
| | Dense-121 | VGG-16 | Inc-v3 | Res-50 | VGG-16 | Inc-v3 |
| MI-TI-DI | 99.8 | 99.8 | 85.7 | 99.4 | 99.2 | 89.1 |
| MI-TI-DI-VT | 100 | 100 | 95.8 | 100 | 100 | 96.0 |
| EMI-TI-DI | 100 | 100 | 93.6 | 100 | 100 | 94.2 |
| MI-TI-DI+RAP-LS | 100 | 100 | 96.9 | 100 | 100 | 97.1 |
| Attack | VGG-16 $\implies$ | | | Inc-v3$\implies$ | | |
| | Res-50 | Dense-121 | Inc-v3 | Res-50 | Dense-121 | VGG-16 |
| MI-TI-DI | 90 | 88.8 | 56.8 | 82.9 | 85.7 | 85.1 |
| MI-TI-DI-VT | 93.9 | 93 | 76.5 | 87.1 | 90.3 | 87.5 |
| EMI-TI-DI | 91.7 | 91.5 | 74.3 | 86 | 88.4 | 86.2 |
| MI-TI-DI+RAP-LS | 97.7 | 97.3 | 81.4 | 90.6 | 93.3 | 91.0 |

Table 9: The targeted attack success rate (%) of VT, EMI, and RAP-LS with the MI-TI-DI baseline.

| Attack | ResNet-50 $\implies$ | | | DenseNet-121$\implies$ | | |
| --- | --- | --- | --- | --- | --- | --- |
| | Dense-121 | VGG-16 | Inc-v3 | Res-50 | VGG-16 | Inc-v3 |
| MI-TI-DI | 74.9 | 62.8 | 10.9 | 44.9 | 38.5 | 7.7 |
| MI-TI-DI-VT | 82.5 | 71.9 | 21.6 | 59.2 | 53.6 | 21.3 |
| EMI-TI-DI | 79.1 | 67.8 | 19.2 | 56.3 | 50.4 | 19.8 |
| MI-TI-DI+RAP-LS | 88.5 | 81.5 | 33.2 | 74.5 | 65.5 | 26.5 |
| Attack | VGG-16 $\implies$ | | | Inc-v3$\implies$ | | |
| | Res-50 | Dense-121 | Inc-v3 | Res-50 | Dense-121 | VGG-16 |
| MI-TI-DI | 11.8 | 13.7 | 0.7 | 1.8 | 4.1 | 2.9 |
| MI-TI-DI-VT | 19.3 | 22.5 | 2.5 | 5.6 | 9.8 | 6.4 |
| EMI-TI-DI | 14.1 | 19.7 | 2.0 | 4.3 | 8.0 | 5.2 |
| MI-TI-DI+RAP-LS | 22.9 | 27.4 | 4.6 | 7.5 | 13.4 | 9.8 |

### C.2 Experimental Results about Ghost Net attack

For combining Ghost net with RAP, we conduct the experiments on our PyTorch codes following the original TensorFlow codes provided by the authors. The main idea of ghost network is to perturb skip connections of ResNet to generate ensemble networks. To achieve this goal, the authors multiply

skip connection by the random scalar $r$ sampled from a uniform distribution. We reimplement this procedure with the hyperparameter about $r$ recommended in the orginal paper on the PyTorch ResNet-50 model.

The targeted attack results (%) are shown in the below Table 10. We also follow the experimental settings in Section 4.1. Here, we use GN to represent Ghost Net method. The results show that ghost network method can improve the transfer attack performance. Combined with RAP-LS, the adversarial transferability can be further improved especially on the inception-v3 model.

Table 10: The targeted attack success rate (%) of GN, and RAP-LS with the MI-TI-DI baseline

| Attack | ResNet-50 $\Longrightarrow$ | | |
| --- | --- | --- | --- |
| | Dense-121 | VGG-16 | Inc-v3 |
| MTDI | 74.9 | 62.8 | 10.9 |
| MTDI-GN | 85.9 | 80.7 | 24.9 |
| MTDI-GN+RAP-LS | 89.6 | 87.7 | 49.7 |

## C.3 Experimental Results about EOT attack

We conducted the experiment of the EOT baseline. We choose the ResNet-50 as the source model and MI-TI-DI as the baseline method. Instead of adding input transformation once like DI, we sample random transformation (resizing and padding) multiple times in each iteration. Then, we add them to following the expectation of transformation (EOT) [1]. We set the number of sampling as 10. Our RAP can be also naturally combined with EOT attack.

The targeted attack results (%) are shown in the below Table 11. We also follow the experimental settings in Section 4.1.

As shown in the below table, **1)** the EOT attack gets a moderate increase on attack performance compared with the baseline MI-TI-DI attack, which demonstrates that EOT could improve adversarial transferability. **2)** Our RAP attack achieves better performance and surpasses the EOT attack by a large margin, especially for Inc-v3 and VGG-16 target models. **3)** Combining RAP with EOT can further boost EOT attack performance. **These results demonstrate that RAP could achieve better adversarial transferability and help find better flat local minima. Besides, the combination of RAP and EOT achieves the best performance among them, which demonstrates that these two methods could complement each other.**

Table 11: The targeted attack success rate (%) of EOT, and RAP-LS with the MI-TI-DI baseline

| Attack | ResNet-50 $\Longrightarrow$ | | |
| --- | --- | --- | --- |
| | Dense-121 | VGG-16 | Inc-v3 |
| MTDI | 74.9 | 62.8 | 10.9 |
| MTDI-EOT | 76.9 | 66.9 | 11.2 |
| MTDI+RAP | 78.2 | 72.9 | 28.3 |
| MTDI-EOT+RAP | 86.1 | 79.5 | 32.8 |

## D Experimental Results about More Defense Models

**The Evaluation on More Defense Models** Here, we show the evaluation of more defense models containing multi-step Adversarial training models in ImageNet [33], Feature Denoising [47], NRP [29], input transformation defense (R&P) [46].

For Feature Denoising, we utilize the pre-trained ResNet-152 model provided by the authors [6]. For AT models on ImageNet, we adopt the pre-trained ResNet-50 AT models provided by the authors [7]. For $\ell_\infty$ norm, we adopt the ResNet-50 AT model with budget $4/255$, which ranks first in the RobustBench leaderboard [8]. For $\ell_2$ norm, we adopt the ResNet-50 AT model with budget $0.5$. The

---

[6] https://github.com/facebookresearch/ImageNet-Adversarial-Training
[7] https://github.com/microsoft/robust-models-transfer
[8] https://robustbench.github.io

untargeted attack performance is shown in Table 12. We follow the experimental settings in Section 4.5 of the main submission. We can observe that our RAP-LS further boosts the transferability of baseline methods on these new defense models, getting a $5.5\%$ boost for the average attack success rate.

For NRP, we adopt the pre-trained purifiers provided by the authors [9]. Since NRP is an offline defense module, we combine it with the two used ensemble AT models and the above two AT models. The untargeted attack performance is shown in Table 13. We also follow the experimental settings in Section 4.5 of the main submission. Combining NRP with AT models is a much stronger defense mechanism, but RAP-LS still achieves an improvement by $0.8\%$.

For R&P, we adopt the source code provided by Dong et al. [7] to implement it. We also combine R&P with the two used ensemble AT models and the two new AT models above. The untargeted attack performance is shown in Table 14. We also follow the experimental settings in Section 4.5 of the main submission. For R&P, RAP-LS achieves an $9.1\%$ increase in terms of average attack success rate.

Table 12: The evaluation of ensemble attacks on **two AT models** and **Feature Noising**.

| Attack | Untarged | | |
| | Res-50 AT ($\ell_2$) | Res-50 AT ($\ell_\infty$) | Feature Denoising |
| --- | --- | --- | --- |
| MTDI | 42.5 | 32.4 | 44.1 |
| MTDI+RAP-LS | **59.5** | **34.4** | **44.4** |
| MTDSI | 56.6 | 35.8 | 45.0 |
| MTDSI+RAP-LS | **70.3** | **36.6** | **45.7** |
| MTDAI | 62.1 | 35.6 | 44.2 |
| MTDAI+RAP-LS | **73.7** | **37.7** | **45.2** |

Table 13: The evaluation of ensemble attacks on defense models with **NRP**.

| Attack | Untarged | | | |
| | Inc-v3$_{adv}$ | IncRes-v2$_{ens}$ | Res-50 AT ($\ell_2$) | Res-50 AT ($\ell_\infty$) |
| --- | --- | --- | --- | --- |
| MTDI | **23.1** | 13.5 | 14.2 | 25.7 |
| MTDI+RAP-LS | 22.7 | **14.8** | **14.9** | **26.3** |
| MTDSI | 22.5 | 14.2 | 15.0 | 26.1 |
| MTDSI+RAP-LS | **24.5** | **15.3** | **15.4** | **26.2** |
| MTDAI | 24.1 | 14.7 | 14.2 | 25.9 |
| MTDAI+RAP-LS | **24.9** | **15.6** | **15.3** | **26.1** |

Table 14: The evaluation of ensemble attacks on defense models with **R&P**.

| Attack | Untarged | | | |
| | Inc-v3$_{adv}$ | IncRes-v2$_{ens}$ | Res-50 AT ($\ell_2$) | Res-50 AT ($\ell_\infty$) |
| --- | --- | --- | --- | --- |
| MTDI | 65.0 | 46.2 | 52.5 | 43.7 |
| MTDI+RAP-LS | **82.1** | **63.2** | **65.3** | **45.8** |
| MTDSI | 86.5 | 69.6 | 64.1 | 45.9 |
| MTDSI+RAP-LS | **93.4** | **84.9** | **74.0** | **46.2** |
| MTDAI | 88.9 | 76.5 | 68.4 | 46.2 |
| MTDAI+RAP-LS | **94.8** | **87.0** | **77.7** | **47.7** |

The above experimental results also show that RAP is less effective when attacking Feature Denoising. We think this is mainly due to the specially designed feature denoising block (*i.e.* the non-local block), and the different settings of maximum perturbation size during adversarial training, as follows.

- Feature Denoising [47] inserts several non-local blocks into network to eliminate the adversarial noise at the feature level. According to [47], for input feature map $F_i$, the non-local block computes a denoised output feature map $F_0$ by taking a weighted average of input features in all spatial locations. Through this, the non-local block would model the global relationship between features in all spatial locations, which may smooth the learned decision

---

boundary. Recalling that our RAP is to boost the transferability by seeking for a flat local minimum. The smoothness of decision boundary could make it harder to escape from certain local minimum, especially for small attack perturbation size, so as to limit the performance improvement of RAP.

- In our experiment, for Feature Denoising, the maximum perturbation size during their training is set to 16/255. In Table 12, the maximum perturbation size of attack is also set to 16/255. The attack size of 16/255 may not be large enough for escaping from local minima for the Feature Denoising model trained with 16/255 perturbation size. In contrast, the maximum perturbation size of AT-$\ell_\infty$ during training is 4/255. To verify this, we conduct an ablation study of increasing the maximum perturbation size to 20/255. Using a larger perturbation size of 20/255, the attacking performance against Feature Denoising is $48.3\%$ for MI-TI-DI and $50.7\%$ for MI-TI-DI+RAP-LS. The relative performance improvement of RAP-LS is $2.4\%$, which is much larger than the relative performance improvement of $0.3\%$ in Table 12 with perturbation size 16/255, which may partially explain the phenomenon.

# E Additional Experimental Results

In this section, we first show the evaluation of targeted attacks with CE loss in Section E.1. Then we show the results of ensemble attacks on more diverse network architectures in Section E.2. In Section E.3, we report the experimental results *w.r.t.* different value of iterations.

## E.1 The Results of Targeted Attacks with CE Loss

Following the settings in main submission, we evaluate the targeted attack performance of the different baseline methods with our method on ResNet-50, DenseNet-121, VGG-16, and Inception-v3. The results of combinational methods are shown in Table 15. The RAP-LS outperforms all combinational methods by a significantly margin. Taking the average attack success rate of all target models as the evaluation metric, RAP-LS achieves $20.9\%$, $18.4\%$, and $15.1\%$ improvements over the MTDI, MTDSI and MTDAI, respectively.

Table 15: The **targeted attack success rate** (%) **of combinational methods with RAP**. The results with $CE$ loss and 400 iterations are reported. The best results are bold and the second best results are underlined.

| Attack | ResNet-50 $\Longrightarrow$ | | | DenseNet-121 $\Longrightarrow$ | | |
|---|---|---|---|---|---|---|
| | Dense-121 | VGG-16 | Inc-v3 | Res-50 | VGG-16 | Inc-v3 |
| MTDI / +RAP / +RAP-LS | 45.5 / 78.3 / **85.9** | 29.8 / 70.5 / **76.7** | 4.5 / 21.3 / **25.3** | 20.0 / 54.0 / **62.7** | 9.9 / 41.7 / **48.7** | 2.6 / 17.5 / **18.5** |
| MTDSI / +RAP / +RAP-LS | 77.7 / 89.0 / **93.7** | 39.9 / 69.4 / **76.7** | 26.9 / 45.3 / **50.8** | 30.5 / 60.4 / **69.5** | 14.9 / 42.8 / **49.7** | 12.7 / 26.6 / **32.5** |
| MTDAI / +RAP / +RAP-LS | 90.2 / 91.4 / **96.1** | 61.8 / 73.7 / **83.4** | 44.5 / 47.9 / **59.0** | 55.8 / 68.4 / **79.3** | 35.1 / 51.8 / **64.1** | 26.3 / 32.4 / **40.4** |

| Attack | VGG-16 $\Longrightarrow$ | | | Inc-v3 $\Longrightarrow$ | | |
|---|---|---|---|---|---|---|
| | Res-50 | Dense-121 | Inc-v3 | Res-50 | Dense-121 | VGG-16 |
| MTDI / +RAP / +RAP-LS | 0.5 / 10.4 / **12.1** | 0.1 / 11.0 / **13.5** | 0.0 / 1.7 / **2.0** | 2.2 / 4.9 / **5.9** | 2.2 / 9.8 / **11.0** | 1.2 / 4.9 / **6.7** |
| MTDSI / +RAP / +RAP-LS | 5.4 / 17.4 / 16.8 | 9.5 / 28.4 / 25.2 | 2.2 / 7.1 / 5.1 | 4.4 / 8.6 / **8.9** | 7.9 / 16.3 / **19.3** | 2.0 / 6.4 / 6.4 |
| MTDAI / +RAP / +RAP-LS | 11.6 / 22.6 / **26.6** | 20.6 / 32.1 / **39.1** | 5.1 / 9.2 / **9.5** | 6.7 / 12.3 / **17.0** | 14.0 / 22.9 / **29.2** | 4.5 / 9.4 / **13.2** |

## E.2 The Results of Ensemble Attacks on Diverse Network Architectures

We also take the evaluation of the ensemble attacks on diverse network architecture (Sec.4.5). We adopt the ensemble-model attack by averaging the logits of different surrogate models, including ResNet-50, DenseNet-121, VGG-16, and Inception-v3. The transfer attack success rate on diverse models are shown in Table 16. Compared with results of single model attack in Table 7, the ensemble attack achieve the better performance. We can observe that our RAP-LS further boosts transferability of the baseline methods on both targeted and untargeted attacks. We take ViT as target model for example. For untargeted attacks, RAP-LS achieves average performance improvements of $19.2\%$. For targeted attacks, RAP-LS achieves average performance improvements of $10.4\%$.

## E.3 The Experimental Results *w.r.t.* Different Value of Iterations

In the main submission, we report the evaluations of $K = 400$. Here, we further report the performance with different values of $K$ for completeness in Table 17 (targeted attack) and Table 18

Table 16: The evaluation of ensemble attacks on diverse network architectures.

| Attack | Untarged | | | Targeted | | |
|---|---|---|---|---|---|---|
| | IncRes-v2 | NASNet-L | ViT-B/16 | IncRes-v2 | NASNet-L | ViT-B/16 |
| MTDI | 98.6 | 99.3 | 46.2 | 65.7 | 80.1 | 2.8 |
| MTDI+RAP-LS | **100** | **100** | **73.2** | **84.4** | **89.7** | **12.7** |
| MTDSI | 99.8 | 100 | 68.3 | 81.7 | 89.4 | 15.0 |
| MTDSI+RAP-LS | **100** | **100** | **85.0** | **89.8** | **92.3** | **25.1** |
| MTDAI | 100 | 100 | 70.7 | 88.8 | 91.2 | 16.8 |
| MTDAI+RAP-LS | **100** | **100** | **84.6** | **90.4** | **91.8** | **27.8** |

(untargeted attack). From the results, we observe that the attacking performance generally increase as $K$ increases for most cases, this is also aligned with prior works [49].

Table 17: The targeted attack success rate (%) of all baseline attacks with our method. The results with logit loss and 10/100/200/300/400 iterations are reported. We highlight the results with $K = 400$ in bold.

| | ResNet-50 → Inception-v3 | | |
|---|---|---|---|
| | Baseline | +RAP | +RAP-LS |
| I | 0.0 / 0.1 / 0.2 / 0.1 / **0.1** | 0.0 / 0.2 / 0.3 / 0.3 / **0.1** | 0.0 / 0.1 / 0.4 / 0.6 / **0.7** |
| MI | 0.1 / 0.1 / 0.2 / 0.1 / **0.1** | 0.0 / 0.6 / 1.0 / 1.0 / **1.1** | 0.1 / 0.1 / 1.4 / 1.6 / **2.4** |
| TI | 0.0 / 0.3 / 0.2 / 0.2 / **0.1** | 0.0 / 0.7 / 0.9 / 1.2 / **0.8** | 0.0 / 0.3 / 1.3 / 1.3 / **1.2** |
| DI | 0.2 / 1.2 / 1.7 / 1.5 / **1.5** | 0.0 / 3.8 / 6.6 / 7.7 / **7.9** | 0.2 / 1.2 / 10.2 / 9.4 / **10.1** |
| SI | 0.3 / 2.6 / 2.4 / 2.0 / **1.8** | 0.2 / 6.6 / 8.2 / 8.6 / **9.3** | 0.3 / 2.6 / 9.6 / 9.3 / **10.5** |
| Admix | 1.4 / 5.7 / 5.9 / 6.0 / **5.8** | 0.6 / 14.6 / 16.6 / 16.5 / **17.1** | 1.4 / 5.7 / 18.5 / 19.2 / **19.6** |
| MI-TI-DI | 1.5 / 7.9 / 9.8 / 10.5 / **10.9** | 0.1 / 12.7 / 22.3 / 26.3 / **28.3** | 1.5 / 7.9 / 26.8 / 30.0 / **33.2** |
| MI-TI-DI-SI | 8.9 / 34.1 / 36.7 / 38.1 / **38.1** | 3.3 / 43.3 / 47.9 / 49.9 / **51.8** | 8.9 / 34.8 / 54.8 / 55.8 / **58.0** |
| MI-TI-DI-Admix | 13.5 / 45.7 / 49.2 / 50.5 / **50.8** | 5.0 / 48.1 / 53.4 / 56.2 / **57.1** | 13.5 / 45.1 / 61.4 / 63.0 / **64.1** |

| | ResNet-50 → DenseNet-121 | | |
|---|---|---|---|
| | Baseline | +RAP | +RAP-LS |
| I | 0.9 / 5.3 / 5.0 / 5.5 / **4.5** | 0.0 / 4.8 / 7.9 / 8.8 / **9.5** | 0.9 / 5.3 / 14.0 / 14.0 / **14.3** |
| MI | 3.4 / 6.3 / 6.3 / 6.0 / **6.3** | 0.2 / 9.0 / 14.1 / 15.8 / **17.5** | 3.4 / 6.3 / 25.9 / 28.9 / **29.6** |
| TI | 2.5 / 8.6 / 8.9 / 9.0 / **7.2** | 0.0 / 7.1 / 10.1 / 11.2 / **11.0** | 2.5 / 8.6 / 16.1 / 16.4 / **17.3** |
| DI | 8.4 / 54.8 / 60.4 / 61.2 / **62.6** | 0.1 / 40.6 / 53.2 / 59.4 / **64.9** | 8.4 / 54.6 / 70.9 / 72.5 / **73.9** |
| SI | 9.7 / 29.6 / 30.4 / 30.4 / **30.0** | 2.2 / 45.8 / 50.9 / 52.5 / **53.2** | 9.7 / 29.6 / 60.0 / 61.1 / **61.1** |
| Admix | 23.6 / 55.6 / 55.5 / 55.6 / **54.6** | 5.3 / 61.2 / 66.0 / 66.9 / **68.0** | 23.6 / 55.6 / 74.4 / 74.7 / **74.6** |
| MI-TI-DI | 16.3 / 66.9 / 71.4 / 73.4 / **74.9** | 1.8 / 56.7 / 71.2 / 76.4 / **78.2** | 16.3 / 66.7 / 85.2 / 85.7 / **88.5** |
| MI-TI-DI-SI | 41.0 / 82.8 / 84.5 / 86.2 / **86.3** | 12.9 / 80.2 / 85.7 / 87.8 / **88.4** | 41.0 / 82.5 / 91.9 / 92.4 / **93.3** |
| MI-TI-DI-Admix | 48.0 / 88.7 / 90.9 / 91.1 / **91.4** | 20.3 / 83.2 / 87.2 / 88.4 / **89.4** | 47.9 / 88.5 / 93.5 / 93.8 / **93.6** |

| | ResNet-50 → VGG-16 | | |
| --- | --- | --- | --- |
| | Baseline | +RAP | +RAP-LS |
| I | 1.0 / 2.7 / 2.6 / 2.3 / **2.4** | 0.0 / 5.6 / 8.3 / 9.8 / **9.8** | 1.0 / 2.7 / 11.4 / 12.8 / **11.8** |
| MI | 1.2 / 2.1 / 2.4 / 2.2 / **2.2** | 0.1 / 8.6 / 12.3 / 14.1 / **14.5** | 1.2 / 2.1 / 18.2 / 20.0 / **20.6** |
| TI | 1.1 / 4.8 / 4.8 / 4.5 / **4.0** | 0.1 / 6.0 / 9.3 / 9.9 / **12.9** | 1.1 / 4.8 / 14.2 / 15.3 / **15.3** |
| DI | 7.6 / 51.0 / 56.9 / 56.6 / **57.2** | 0.4 / 42.4 / 55.0 / 61.5 / **63.4** | 7.6 / 51.0 / 69.3 / 69.8 / **69.3** |
| SI | 4.4 / 10.4 / 8.9 / 8.8 / **9.5** | 1.1 / 27.8 / 31.1 / 30.7 / **32.8** | 4.4 / 10.4 / 35.8 / 35.1 / **36.0** |
| Admix | 10.6 / 24.9 / 25.0 / 26.2 / **26.0** | 3.6 / 41.4 / 45.2 / 43.8 / **45.4** | 10.6 / 24.9 / 51.7 / 51.9 / **51.6** |
| MI-TI-DI | 12.1 / 55.9 / 61.0 / 63.9 / **62.8** | 1.5 / 53.0 / 64.7 / 70.9 / **72.9** | 12.1 / 55.8 / 78.5 / 81.7 / **81.5** |
| MI-TI-DI-SI | 24.6 / 67.4 / 68.5 / 69.7 / **70.1** | 8.2 / 66.4 / 73.7 / 75.2 / **77.7** | 24.5 / 66.4 / 82.4 / 83.7 / **84.7** |
| MI-TI-DI-Admix | 33.4 / 75.3 / 77.5 / 78.7 / **79.9** | 14.6 / 70.4 / 76.7 / 78.3 / **79.0** | 33.3 / 75.2 / 85.4 / 86.4 / **86.3** |

| | DenseNet121 → Inception-v3 | | |
| --- | --- | --- | --- |
| | Baseline | +RAP | +RAP-LS |
| I | 0.0 / 0.1 / 0.2 / 0.1 / **0.0** | 0.0 / 0.6 / 0.9 / 0.7 / **0.8** | 0.0 / 0.1 / 1.0 / 1.3 / **1.2** |
| MI | 0.2 / 0.2 / 0.3 / 0.3 / **0.3** | 0.0 / 1.2 / 2.1 / 2.1 / **2.0** | 0.2 / 0.2 / 2.5 / 3.7 / **3.4** |
| TI | 0.0 / 0.4 / 0.3 / 0.5 / **0.2** | 0.0 / 1.2 / 1.5 / 1.6 / **2.1** | 0.0 / 0.4 / 2.6 / 3.1 / **3.0** |
| DI | 0.3 / 1.9 / 1.4 / 1.7 / **1.4** | 0.0 / 4.1 / 7.0 / 7.6 / **8.8** | 0.3 / 1.9 / 9.3 / 9.9 / **10.0** |
| SI | 0.3 / 1.5 / 1.8 / 1.6 / **1.6** | 0.1 / 7.6 / 9.2 / 10.0 / **8.5** | 0.3 / 1.5 / 9.2 / 10.7 / **10.4** |
| Admix | 1.7 / 5.0 / 5.4 / 5.5 / **5.0** | 0.2 / 15.8 / 17.0 / 17.7 / **17.1** | 1.7 / 5.0 / 18.5 / 18.2 / **17.6** |
| MI-TI-DI | 1.2 / 6.8 / 7.9 / 8.7 / **7.7** | 0.1 / 13.0 / 19.7 / 22.2 / **23.0** | 1.2 / 6.7 / 21.9 / 26.2 / **26.5** |
| MI-TI-DI-SI | 5.1 / 17.6 / 18.9 / 19.3 / **19.8** | 2.0 / 30.4 / 35.1 / 37.0 / **39.0** | 5.2 / 17.7 / 36.8 / 38.9 / **39.2** |
| MI-TI-DI-Admix | 11.4 / 30.5 / 32.2 / 31.4 / **32.0** | 3.9 / 36.7 / 41.3 / 42.2 / **43.5** | 11.2 / 31.2 / 47.2 / 49.2 / **49.3** |

| | DenseNet121 → ResNet-50 | | |
| --- | --- | --- | --- |
| | Baseline | +RAP | +RAP-LS |
| I | 1.8 / 6.5 / 5.6 / 5.5 / **5.0** | 0.2 / 7.7 / 11.2 / 12.4 / **12.8** | 1.8 / 6.5 / 18.7 / 19.0 / **17.9** |
| MI | 3.4 / 5.4 / 5.2 / 4.9 / **4.6** | 0.3 / 10.2 / 14.3 / 16.3 / **16.2** | 3.4 / 5.4 / 23.6 / 26.3 / **26.5** |
| TI | 2.6 / 8.1 / 7.9 / 8.4 / **8.4** | 0.2 / 7.8 / 10.9 / 12.1 / **13.5** | 2.6 / 8.1 / 19.2 / 20.2 / **20.8** |
| DI | 6.3 / 30.4 / 33.1 / 32.0 / **30.2** | 0.4 / 33.6 / 44.1 / 48.7 / **52.6** | 6.3 / 30.8 / 58.8 / 60.4 / **60.4** |
| SI | 7.3 / 16.5 / 15.9 / 14.8 / **14.2** | 1.5 / 33.8 / 39.5 / 41.4 / **41.5** | 7.3 / 16.5 / 44.7 / 44.8 / **43.4** |
| Admix | 16.4 / 32.6 / 30.3 / 28.8 / **29.3** | 3.7 / 48.3 / 52.9 / 53.4 / **53.0** | 16.4 / 32.6 / 60.1 / 58.8 / **58.2** |
| MI-TI-DI | 8.3 / 40.3 / 44.6 / 46.3 / **44.9** | 0.9 / 42.0 / 56.4 / 62.4 / **64.3** | 8.3 / 40.1 / 69.5 / 72.8 / **74.5** |
| MI-TI-DI-SI | 18.6 / 52.3 / 54.1 / 56.2 / **55.0** | 6.6 / 60.3 / 67.5 / 70.6 / **71.2** | 18.6 / 52.5 / 73.8 / 75.5 / **75.8** |
| MI-TI-DI-Admix | 27.6 / 66.3 / 69.7 / 69.8 / **69.1** | 12.1 / 66.4 / 70.8 / 73.2 / **74.2** | 27.6 / 66.4 / 81.4 / 82.0 / **82.1** |

| | DenseNet121 → VGG-16 | | |
| --- | --- | --- | --- |
| | Baseline | +RAP | +RAP-LS |
| I | 0.6 / 3.8 / 3.5 / 3.5 / **2.9** | 0.1 / 6.2 / 9.3 / 10.5 / **10.1** | 0.6 / 3.8 / 14.5 / 15.7 / **15.9** |
| MI | 1.6 / 2.4 / 2.6 / 2.7 / **3.1** | 0.2 / 8.6 / 12.2 / 13.0 / **13.4** | 1.6 / 2.4 / 19.5 / 21.7 / **23.2** |
| TI | 1.1 / 5.6 / 5.8 / 4.8 / **5.2** | 0.1 / 6.3 / 9.1 / 11.0 / **12.4** | 1.1 / 5.6 / 16.5 / 17.0 / **16.4** |
| DI | 4.1 / 29.8 / 32.7 / 33.1 / **32.1** | 0.2 / 31.5 / 44.7 / 48.7 / **49.5** | 4.1 / 29.9 / 57.2 / 56.5 / **58.9** |
| SI | 2.8 / 9.8 / 8.8 / 8.5 / **8.4** | 0.6 / 25.8 / 28.2 / 31.4 / **31.0** | 2.8 / 9.8 / 33.5 / 35.3 / **35.2** |
| Admix | 10.2 / 23.3 / 22.1 / 21.3 / **21.5** | 1.7 / 39.4 / 42.2 / 43.0 / **42.7** | 10.2 / 23.3 / 49.7 / 49.3 / **48.2** |
| MI-TI-DI | 6.1 / 32.4 / 36.3 / 39.0 / **38.5** | 0.7 / 36.2 / 49.9 / 53.2 / **55.0** | 6.1 / 32.6 / 61.8 / 64.3 / **65.5** |
| MI-TI-DI-SI | 12.4 / 40.2 / 41.9 / 42.2 / **42.0** | 4.6 / 46.9 / 54.0 / 57.0 / **58.4** | 12.4 / 40.0 / 61.3 / 62.4 / **62.3** |
| MI-TI-DI-Admix | 20.0 / 53.2 / 55.0 / 55.7 / **54.7** | 9.4 / 54.8 / 60.1 / 61.8 / **63.1** | 19.9 / 53.1 / 68.1 / 69.7 / **69.3** |

| | VGG-16 → Inception-v3 | | |
| --- | --- | --- | --- |
| | Baseline | +RAP | +RAP-LS |
| I | 0.0 / 0.0 / 0.0 / 0.0 / **0.0** | 0.0 / 0.1 / 0.0 / 0.1 / **0.1** | 0.0 / 0.0 / 0.2 / 0.0 / **0.2** |
| MI | 0.0 / 0.0 / 0.0 / 0.0 / **0.0** | 0.0 / 0.0 / 0.2 / 0.0 / **0.0** | 0.0 / 0.0 / 0.2 / 0.5 / **0.3** |
| TI | 0.0 / 0.0 / 0.0 / 0.1 / **0.0** | 0.0 / 0.1 / 0.1 / 0.1 / **0.1** | 0.0 / 0.0 / 0.4 / 0.4 / **0.4** |
| DI | 0.0 / 0.0 / 0.0 / 0.0 / **0.0** | 0.0 / 0.0 / 0.4 / 0.6 / **0.4** | 0.0 / 0.0 / 0.7 / 0.7 / **1.1** |
| SI | 0.0 / 0.4 / 0.3 / 0.2 / **0.2** | 0.0 / 2.0 / 1.5 / 2.0 / **1.7** | 0.0 / 0.6 / 1.6 / 1.9 / **1.8** |
| Admix | 0.1 / 0.7 / 0.8 / 0.6 / **0.7** | 0.0 / 2.7 / 2.2 / 2.3 / **2.4** | 0.1 / 1.0 / 2.3 / 3.0 / **2.8** |
| MI-TI-DI | 0.1 / 1.0 / 0.8 / 1.1 / **0.7** | 0.0 / 1.8 / 2.4 / 3.0 / **3.4** | 0.1 / 0.9 / 3.4 / 4.0 / **4.6** |
| MI-TI-DI-SI | 1.7 / 7.7 / 9.1 / 9.8 / **9.6** | 0.6 / 12.2 / 14.5 / 13.8 / **15.2** | 1.7 / 8.6 / 11.4 / 12.1 / **13.7** |
| MI-TI-DI-Admix | 3.6 / 12.4 / 12.2 / 11.5 / **11.6** | 1.1 / 14.5 / 16.1 / 15.9 / **17.1** | 3.4 / 11.2 / 15.9 / 17.4 / **17.6** |

| | VGG-16 → ResNet-50 | | |
| --- | --- | --- | --- |
| | Baseline | +RAP | +RAP-LS |
| I | 0.2 / 0.4 / 0.3 / 0.3 / **0.1** | 0.0 / 1.0 / 0.8 / 0.8 / **0.7** | 0.2 / 0.5 / 1.4 / 1.5 / **1.4** |
| MI | 0.4 / 0.5 / 0.6 / 0.5 / **0.5** | 0.2 / 1.1 / 1.3 / 1.3 / **1.3** | 0.4 / 0.2 / 2.1 / 2.4 / **1.9** |
| TI | 0.3 / 1.0 / 0.7 / 0.9 / **0.7** | 0.0 / 1.4 / 1.5 / 1.4 / **1.2** | 0.3 / 1.0 / 3.0 / 3.3 / **3.2** |
| DI | 0.5 / 2.8 / 3.1 / 3.4 / **2.8** | 0.0 / 4.9 / 6.7 / 6.5 / **7.3** | 0.5 / 3.9 / 9.5 / 10.1 / **9.7** |
| SI | 1.4 / 4.4 / 3.9 / 3.8 / **3.3** | 0.4 / 9.2 / 9.0 / 9.1 / **9.8** | 1.4 / 4.3 / 10.1 / 9.4 / **9.8** |
| Admix | 4.6 / 7.3 / 6.7 / 5.8 / **5.6** | 0.7 / 10.6 / 11.3 / 10.9 / **11.1** | 4.7 / 7.3 / 11.6 / 12.5 / **11.9** |
| MI-TI-DI | 1.8 / 10.2 / 11.7 / 11.9 / **11.8** | 0.0 / 10.8 / 14.6 / 15.7 / **16.7** | 1.8 / 9.5 / 20.2 / 21.6 / **22.9** |
| MI-TI-DI-SI | 8.8 / 30.1 / 31.6 / 30.3 / **31.0** | 3.2 / 30.8 / 32.5 / 33.5 / **35.3** | 9.0 / 29.5 / 36.9 / 38.5 / **38.7** |
| MI-TI-DI-Admix | 15.2 / 34.6 / 35.1 / 36.6 / **36.2** | 5.4 / 34.7 / 37.3 / 38.1 / **39.0** | 15.3 / 35.5 / 43.2 / 42.9 / **43.1** |

| | VGG-16 → DenseNet-121 | | |
| --- | --- | --- | --- |
| | Baseline | +RAP | +RAP-LS |
| I | 0.1 / 0.2 / 0.4 / 0.3 / **0.2** | 0.0 / 0.7 / 1.1 / 0.7 / **1.4** | 0.1 / 0.3 / 1.2 / 1.5 / **1.7** |
| MI | 0.3 / 0.8 / 0.6 / 0.6 / **0.5** | 0.0 / 1.1 / 1.4 / 2.1 / **2.3** | 0.3 / 0.6 / 2.4 / 3.2 / **3.0** |
| TI | 0.1 / 0.6 / 1.1 / 1.0 / **0.8** | 0.0 / 0.9 / 1.7 / 1.6 / **1.7** | 0.1 / 0.9 / 2.5 / 2.7 / **2.9** |
| DI | 0.2 / 3.8 / 4.8 / 4.1 / **3.8** | 0.0 / 5.0 / 7.6 / 7.8 / **8.4** | 0.2 / 3.7 / 11.9 / 12.2 / **12.7** |
| SI | 1.3 / 9.0 / 8.9 / 7.7 / **7.2** | 0.3 / 14.0 / 15.6 / 16.4 / **16.8** | 1.3 / 8.2 / 17.0 / 17.4 / **17.8** |
| Admix | 4.9 / 14.3 / 13.4 / 13.2 / **13.0** | 0.7 / 17.9 / 20.5 / 20.2 / **20.2** | 4.9 / 14.0 / 23.9 / 24.2 / **23.6** |
| MI-TI-DI | 1.5 / 12.1 / 13.4 / 13.9 / **13.7** | 0.1 / 9.7 / 15.7 / 17.4 / **19.4** | 1.6 / 12.1 / 24.4 / 26.3 / **27.4** |
| MI-TI-DI-SI | 13.0 / 38.9 / 41.5 / 42.8 / **41.7** | 3.8 / 37.8 / 42.0 / 43.8 / **44.4** | 12.8 / 37.3 / 48.6 / 49.8 / **49.6** |
| MI-TI-DI-Admix | 19.0 / 45.5 / 47.0 / 47.7 / **48.0** | 6.8 / 41.3 / 45.2 / 44.8 / **45.1** | 19.1 / 45.3 / 52.9 / 54.9 / **55.2** |

| | Inc-v3 → ResNet-50 | | |
| --- | --- | --- | --- |
| | Baseline | +RAP | +RAP-LS |
| I | 0.2 / 0.4 / 0.3 / 0.1 / **0.2** | 0.0 / 0.2 / 0.7 / 0.6 / **0.9** | 0.2 / 0.4 / 1.0 / 0.7 / **0.5** |
| MI | 0.1 / 0.3 / 0.3 / 0.2 / **0.2** | 0.0 / 0.6 / 1.4 / 1.5 / **1.7** | 0.1 / 0.3 / 0.8 / 1.6 / **1.5** |
| TI | 0.2 / 0.3 / 0.2 / 0.2 / **0.2** | 0.0 / 0.2 / 0.6 / 0.9 / **0.5** | 0.2 / 0.3 / 1.0 / 0.7 / **0.7** |
| DI | 0.2 / 1.5 / 1.4 / 1.9 / **1.6** | 0.1 / 2.5 / 4.3 / 4.3 / **4.6** | 0.2 / 1.5 / 5.0 / 5.1 / **6.4** |
| SI | 0.3 / 0.3 / 0.3 / 0.6 / **0.6** | 0.4 / 1.9 / 2.6 / 2.6 / **2.9** | 0.3 / 0.3 / 2.4 / 2.8 / **2.5** |
| Admix | 1.2 / 1.9 / 2.2 / 1.9 / **1.5** | 0.6 / 5.0 / 4.9 / 5.2 / **4.9** | 1.2 / 1.9 / 5.7 / 5.7 / **5.2** |
| MI-TI-DI | 0.6 / 1.6 / 2.0 / 2.4 / **1.8** | 0.0 / 4.2 / 6.3 / 7.7 / **8.3** | 0.6 / 1.7 / 6.2 / 7.0 / **7.5** |
| MI-TI-DI-SI | 1.5 / 4.7 / 5.5 / 5.8 / **5.6** | 0.7 / 8.6 / 10.3 / 11.1 / **11.9** | 1.5 / 5.0 / 10.0 / 9.6 / **10.7** |
| MI-TI-DI-Admix | 2.8 / 8.9 / 9.5 / 9.6 / **9.6** | 1.4 / 12.6 / 14.0 / 13.6 / **13.6** | 2.8 / 8.6 / 14.5 / 15.1 / **16.7** |

| | Inc-v3 → DenseNet-121 | | |
| --- | --- | --- | --- |
| | Baseline | +RAP | +RAP-LS |
| I | 0.0 / 0.0 / 0.2 / 0.0 / **0.2** | 0.0 / 0.2 / 0.4 / 0.6 / **0.6** | 0.0 / 0.0 / 0.2 / 0.4 / **0.3** |
| MI | 0.0 / 0.1 / 0.2 / 0.1 / **0.1** | 0.1 / 0.7 / 1.0 / 1.1 / **1.6** | 0.0 / 0.1 / 1.0 / 1.1 / **1.5** |
| TI | 0.0 / 0.3 / 0.2 / 0.0 / **0.1** | 0.0 / 0.3 / 0.3 / 0.3 / **0.7** | 0.0 / 0.3 / 0.9 / 0.9 / **0.6** |
| DI | 0.1 / 1.3 / 2.5 / 3.0 / **2.8** | 0.0 / 2.7 / 4.4 / 5.4 / **5.8** | 0.1 / 1.3 / 5.9 / 7.0 / **7.5** |
| SI | 0.2 / 0.7 / 0.9 / 0.8 / **0.9** | 0.0 / 2.4 / 3.3 / 2.9 / **2.7** | 0.2 / 0.7 / 3.2 / 3.1 / **3.2** |
| Admix | 1.1 / 2.6 / 2.5 / 2.3 / **2.0** | 0.5 / 7.2 / 7.7 / 7.0 / **6.9** | 1.1 / 2.6 / 8.2 / 7.3 / **7.5** |
| MI-TI-DI | 0.5 / 3.1 / 3.8 / 4.5 / **4.1** | 0.2 / 5.4 / 10.8 / 12.6 / **14.8** | 0.5 / 3.3 / 10.6 / 11.8 / **13.4** |
| MI-TI-DI-SI | 1.9 / 9.0 / 9.4 / 9.5 / **10.4** | 1.1 / 15.5 / 19.8 / 19.8 / **21.2** | 1.9 / 9.0 / 19.1 / 20.2 / **20.9** |
| MI-TI-DI-Admix | 4.6 / 15.7 / 16.8 / 17.4 / **17.9** | 2.4 / 23.2 / 24.5 / 26.6 / **27.5** | 4.6 / 15.0 / 29.1 / 30.2 / **31.6** |

| | Inc-v3 → VGG-16 | | |
| --- | --- | --- | --- |
| | Baseline | +RAP | +RAP-LS |
| I | 0.0 / 0.3 / 0.1 / 0.1 / **0.1** | 0.0 / 0.2 / 0.8 / 0.6 / **0.5** | 0.0 / 0.3 / 0.2 / 0.5 / **0.5** |
| MI | 0.1 / 0.1 / 0.2 / 0.2 / **0.2** | 0.1 / 0.4 / 0.8 / 1.2 / **1.3** | 0.1 / 0.1 / 0.4 / 0.8 / **1.0** |
| TI | 0.1 / 0.2 / 0.2 / 0.1 / **0.2** | 0.1 / 0.4 / 0.5 / 0.6 / **0.8** | 0.1 / 0.2 / 0.6 / 0.6 / **0.6** |
| DI | 0.3 / 2.0 / 2.8 / 2.3 / **2.6** | 0.1 / 1.8 / 4.3 / 5.2 / **6.3** | 0.3 / 2.0 / 6.8 / 7.3 / **8.1** |
| SI | 0.0 / 0.7 / 0.6 / 0.4 / **0.5** | 0.2 / 2.0 / 1.5 / 1.5 / **1.5** | 0.0 / 0.7 / 1.6 / 2.3 / **2.3** |
| Admix | 0.5 / 1.6 / 1.0 / 1.0 / **1.3** | 0.4 / 3.2 / 3.8 / 4.1 / **3.3** | 0.5 / 1.6 / 4.5 / 3.8 / **4.4** |
| MI-TI-DI | 0.3 / 2.0 / 2.4 / 2.7 / **2.9** | 0.1 / 3.8 / 6.5 / 7.3 / **8.0** | 0.3 / 2.0 / 8.0 / 7.9 / **9.8** |
| MI-TI-DI-SI | 0.7 / 3.7 / 3.6 / 4.1 / **4.2** | 0.5 / 7.6 / 7.5 / 8.5 / **8.9** | 0.7 / 3.2 / 6.7 / 8.1 / **8.6** |
| MI-TI-DI-Admix | 2.3 / 6.9 / 8.3 / 8.6 / **8.4** | 1.3 / 10.5 / 12.5 / 12.0 / **12.0** | 2.3 / 7.0 / 11.9 / 12.8 / **12.1** |

Table 18: The untargeted attack success rate (%) of all baseline attacks with RAP. The results with $CE$ loss and 10/100/200/300/400 iterations are reported. We highlight the results with $K = 400$ in bold.

| | ResNet-50 → Inception-v3 | | |
| | Baseline | +RAP | +RAP-LS |
|---|---|---|---|
| I | 25.9 / 35.5 / 35.3 / 34.7 / **34.6** | 12.3 / 48.3 / 54.1 / 55.5 / **57.0** | 25.7 / 36.0 / 54.1 / 56.5 / **57.2** |
| MI | 53.2 / 50.7 / 51.0 / 50.6 / **50.3** | 26.2 / 58.7 / 68.9 / 73.4 / **75.9** | 53.2 / 50.7 / 64.3 / 73.6 / **77.4** |
| TI | 30.0 / 45.3 / 44.0 / 45.3 / **45.5** | 16.4 / 57.9 / 63.9 / 64.6 / **66.1** | 30.0 / 45.1 / 62.3 / 65.3 / **67.0** |
| DI | 46.0 / 60.5 / 59.5 / 59.4 / **57.7** | 27.3 / 80.7 / 82.8 / 83.4 / **82.9** | 46.0 / 61.0 / 86.0 / 85.7 / **85.0** |
| SI | 50.1 / 66.0 / 65.6 / 66.0 / **65.9** | 60.6 / 80.5 / 80.9 / 80.9 / **79.7** | 49.9 / 66.6 / 85.2 / 85.0 / **84.4** |
| Admix | 66.6 / 78.7 / 79.2 / 78.0 / **77.7** | 73.9 / 87.6 / 87.0 / 86.8 / **87.4** | 67.6 / 79.4 / 91.8 / 92.3 / **92.6** |
| MI-TI-DI | 82.1 / 85.8 / 86.4 / 85.9 / **85.7** | 61.9 / 93.9 / 95.3 / 95.6 / **96.0** | 82.1 / 85.8 / 95.9 / 96.4 / **96.9** |
| MI-TI-DI-SI | 94.2 / 96.8 / 97.2 / 97.0 / **97.0** | 92.3 / 98.9 / 98.9 / 99.0 / **99.1** | 94.2 / 96.7 / 99.0 / 99.3 / **99.1** |
| MI-TI-DI-Admix | 97.3 / 98.6 / 98.5 / 98.5 / **98.3** | 95.1 / 99.4 / 99.4 / 99.3 / **99.2** | 97.3 / 98.5 / 99.8 / 99.8 / **99.8** |

| | ResNet-50 → DenseNet-121 | | |
| | Baseline | +RAP | +RAP-LS |
|---|---|---|---|
| I | 67.4 / 79.9 / 79.1 / 79.0 / **79.2** | 26.7 / 84.8 / 91.1 / 90.8 / **91.5** | 67.8 / 80.1 / 89.8 / 91.3 / **91.9** |
| MI | 87.3 / 85.4 / 86.4 / 85.9 / **85.8** | 45.2 / 85.3 / 91.3 / 93.9 / **95.0** | 87.3 / 85.4 / 90.8 / 95.0 / **96.1** |
| TI | 73.2 / 83.0 / 82.2 / 81.6 / **82.0** | 30.9 / 87.3 / 91.5 / 93.3 / **94.1** | 72.9 / 82.4 / 90.9 / 94.2 / **95.1** |
| DI | 92.8 / 98.9 / 99.2 / 99.0 / **99.0** | 52.6 / 99.0 / 99.6 / 99.7 / **99.6** | 92.8 / 99.0 / 99.6 / 99.7 / **99.7** |
| SI | 89.1 / 95.7 / 95.6 / 95.3 / **94.9** | 91.3 / 98.9 / 99.0 / 99.2 / **98.9** | 89.1 / 95.7 / 99.7 / 99.7 / **99.7** |
| Admix | 96.6 / 98.9 / 98.5 / 98.1 / **97.9** | 96.2 / 99.6 / 99.6 / 99.6 / **99.6** | 96.4 / 98.5 / 99.9 / 99.9 / **99.9** |
| MI-TI-DI | 98.2 / 99.7 / 99.8 / 99.8 / **99.8** | 86.4 / 99.9 / 100 / 100 / **100** | 98.2 / 99.7 / 99.9 / 100 / **100** |
| MI-TI-DI-SI | 99.8 / 100 / 100 / 100 / **100** | 98.8 / 100 / 100 / 100 / **100** | 99.8 / 100 / 100 / 100 / **100** |
| MI-TI-DI-Admix | 99.9 / 100 / 100 / 100 / **100** | 99.5 / 100 / 100 / 100 / **100** | 99.9 / 100 / 100 / 100 / **100** |

| | ResNet-50 → VGG-16 | | |
| | Baseline | +RAP | +RAP-LS |
|---|---|---|---|
| I | 68.2 / 77.4 / 78.1 / 77.4 / **78.0** | 36.2 / 84.6 / 89.2 / 90.7 / **91.1** | 68.4 / 77.3 / 87.1 / 90.9 / **92.9** |
| MI | 82.5 / 82.8 / 82.9 / 82.7 / **82.4** | 53.1 / 85.5 / 92.2 / 93.1 / **93.9** | 82.5 / 82.8 / 89.3 / 93.7 / **94.5** |
| TI | 70.6 / 80.5 / 79.8 / 80.8 / **81.0** | 39.3 / 86.9 / 90.6 / 92.5 / **93.1** | 71.1 / 80.0 / 89.0 / 91.9 / **93.3** |
| DI | 92.3 / 99.1 / 99.1 / 99.0 / **99.0** | 64.4 / 99.4 / 99.7 / 99.7 / **99.6** | 92.3 / 99.1 / 99.8 / 99.9 / **99.7** |
| SI | 82.2 / 90.0 / 88.9 / 89.6 / **88.6** | 81.3 / 95.7 / 95.8 / 95.7 / **95.7** | 82.1 / 89.3 / 97.7 / 97.8 / **97.2** |
| Admix | 92.3 / 95.4 / 96.0 / 95.6 / **95.8** | 91.6 / 97.9 / 98.4 / 97.8 / **97.7** | 92.7 / 95.9 / 98.9 / 99.0 / **99.0** |
| MI-TI-DI | 97.9 / 99.7 / 99.7 / 99.8 / **99.8** | 85.9 / 99.5 / 100 / 100 / **100** | 97.9 / 99.7 / 99.9 / 99.9 / **99.9** |
| MI-TI-DI-SI | 99.1 / 99.8 / 99.8 / 99.7 / **99.7** | 97.4 / 99.7 / 99.9 / 99.9 / **99.9** | 99.1 / 99.8 / 99.8 / 99.8 / **99.8** |
| MI-TI-DI-Admix | 99.2 / 99.8 / 99.8 / 99.8 / **99.8** | 98.5 / 99.7 / 99.9 / 99.9 / **99.9** | 99.2 / 99.8 / 99.9 / 99.9 / **99.9** |

| | DenseNet-121 → Inception-v3 | | |
| | Baseline | +RAP | +RAP-LS |
|---|---|---|---|
| I | 31.2 / 48.5 / 46.9 / 46.3 / **46.5** | 18.0 / 54.9 / 58.1 / 59.8 / **60.2** | 31.6 / 46.9 / 58.9 / 61.0 / **61.1** |
| MI | 56.8 / 58.8 / 59.3 / 60.6 / **59.3** | 32.2 / 65.6 / 74.1 / 78.9 / **80.4** | 56.8 / 58.8 / 74.6 / 80.0 / **82.8** |
| TI | 37.7 / 54.0 / 55.1 / 54.6 / **54.2** | 20.4 / 61.0 / 64.7 / 67.3 / **66.7** | 38.2 / 54.5 / 65.4 / 67.6 / **70.0** |
| DI | 51.0 / 67.9 / 68.3 / 66.7 / **67.6** | 31.4 / 84.0 / 86.8 / 86.7 / **86.6** | 51.0 / 68.0 / 89.0 / 88.8 / **86.9** |
| SI | 54.7 / 71.5 / 71.6 / 70.3 / **71.6** | 61.1 / 82.9 / 83.1 / 83.5 / **83.2** | 53.9 / 71.0 / 86.4 / 87.0 / **87.4** |
| Admix | 72.5 / 82.0 / 82.6 / 82.2 / **82.0** | 73.0 / 89.9 / 90.3 / 89.5 / **89.8** | 71.7 / 82.8 / 93.9 / 93.2 / **93.8** |
| MI-TI-DI | 81.5 / 89.7 / 89.8 / 89.4 / **89.1** | 62.5 / 94.8 / 96.8 / 97.1 / **97.1** | 81.5 / 89.6 / 96.1 / 96.9 / **97.1** |
| MI-TI-DI-SI | 92.3 / 95.2 / 94.9 / 95.1 / **95.1** | 88.6 / 97.7 / 98.0 / 98.0 / **98.3** | 92.4 / 95.2 / 97.8 / 98.5 / **98.4** |
| MI-TI-DI-Admix | 95.8 / 97.7 / 97.2 / 97.3 / **97.9** | 93.2 / 98.6 / 98.6 / 99.0 / **98.8** | 95.4 / 97.6 / 99.0 / 98.9 / **98.9** |

| | DenseNet-121 → ResNet-50 | | |
| | Baseline | +RAP | +RAP-LS |
|---|---|---|---|
| I | 76.1 / 88.0 / 87.5 / 87.1 / **87.4** | 35.7 / 90.1 / 93.5 / 93.2 / **94.2** | 76.1 / 88.0 / 91.2 / 92.9 / **94.3** |
| MI | 87.7 / 90.5 / 91.2 / 90.8 / **90.3** | 55.6 / 91.1 / 96.2 / 96.9 / **97.6** | 87.7 / 90.5 / 95.4 / 97.2 / **97.9** |
| TI | 79.2 / 90.4 / 90.0 / 89.9 / **89.6** | 36.9 / 90.1 / 93.2 / 95.0 / **94.2** | 79.0 / 89.8 / 92.7 / 94.3 / **94.8** |
| DI | 91.1 / 98.0 / 98.3 / 98.2 / **98.2** | 57.0 / 98.6 / 99.3 / 99.7 / **99.6** | 91.1 / 98.0 / 99.5 / 99.6 / **99.7** |
| SI | 89.6 / 95.2 / 94.8 / 95.3 / **95.1** | 83.0 / 96.5 / 96.7 / 96.3 / **96.9** | 89.4 / 95.0 / 98.7 / 98.8 / **98.8** |
| Admix | 96.3 / 97.6 / 97.7 / 97.7 / **97.0** | 90.9 / 98.8 / 98.8 / 99.0 / **99.0** | 95.7 / 97.9 / 99.3 / 99.2 / **99.2** |
| MI-TI-DI | 96.3 / 99.3 / 99.5 / 99.4 / **99.4** | 84.4 / 99.2 / 99.8 / 99.8 / **99.8** | 96.3 / 99.2 / 99.8 / 99.9 / **100** |
| MI-TI-DI-SI | 98.3 / 99.7 / 99.8 / 99.8 / **99.8** | 95.8 / 99.7 / 99.9 / 99.9 / **99.9** | 98.3 / 99.7 / 99.9 / 99.9 / **99.9** |
| MI-TI-DI-Admix | 99.2 / 99.7 / 99.8 / 99.8 / **99.8** | 97.9 / 99.9 / 99.8 / 99.8 / **99.8** | 99.0 / 99.7 / 99.9 / 99.9 / **99.9** |

|  | DenseNet-121 → VGG-16 | | |
|---|---|---|---|
|  | Baseline | +RAP | +RAP-LS |
| I | 75.1 / 84.7 / 85.2 / 84.9 / **85.1** | 42.2 / 87.5 / 90.7 / 91.2 / **91.7** | 75.1 / 84.6 / 89.2 / 91.7 / **92.8** |
| MI | 85.1 / 87.2 / 88.6 / 87.9 / **87.5** | 58.4 / 90.2 / 93.7 / 95.1 / **96.0** | 85.1 / 87.2 / 94.2 / 97.0 / **97.6** |
| TI | 74.4 / 86.3 / 86.4 / 87.3 / **87.0** | 44.2 / 87.8 / 89.6 / 91.0 / **92.1** | 74.5 / 85.8 / 90.3 / 92.2 / **93.3** |
| DI | 90.8 / 98.0 / 98.4 / 98.1 / **98.1** | 63.3 / 98.6 / 99.2 / 99.6 / **99.4** | 90.8 / 97.9 / 99.4 / 99.2 / **99.4** |
| SI | 84.2 / 91.5 / 91.4 / 91.4 / **91.9** | 78.5 / 93.9 / 94.5 / 95.2 / **95.0** | 83.9 / 91.6 / 96.9 / 97.1 / **97.5** |
| Admix | 93.5 / 95.7 / 96.0 / 96.1 / **95.6** | 87.8 / 97.4 / 97.5 / 97.6 / **97.7** | 92.0 / 96.1 / 98.9 / 98.7 / **98.6** |
| MI-TI-DI | 95.1 / 99.0 / 99.2 / 99.2 / **99.2** | 84.2 / 99.1 / 99.4 / 99.5 / **99.5** | 95.1 / 99.0 / 99.9 / 100 / **100** |
| MI-TI-DI-SI | 97.9 / 99.5 / 99.4 / 99.4 / **99.2** | 93.3 / 99.0 / 99.2 / 99.3 / **99.3** | 97.9 / 99.4 / 99.7 / 99.7 / **99.7** |
| MI-TI-DI-Admix | 98.4 / 99.4 / 99.4 / 99.5 / **99.4** | 96.1 / 99.7 / 99.7 / 99.6 / **99.6** | 98.3 / 99.4 / 99.8 / 99.7 / **99.8** |

|  | VGG-16 → Inception-v3 | | |
|---|---|---|---|
|  | Baseline | +RAP | +RAP-LS |
| I | 14.3 / 22.2 / 22.0 / 22.2 / **22.0** | 9.4 / 23.8 / 26.1 / 23.7 / **24.7** | 14.4 / 21.8 / 24.1 / 25.4 / **24.9** |
| MI | 32.3 / 31.3 / 31.0 / 30.1 / **30.0** | 16.4 / 30.4 / 36.9 / 42.0 / **42.7** | 32.4 / 30.7 / 35.0 / 39.2 / **42.2** |
| TI | 18.7 / 30.2 / 29.6 / 29.7 / **29.1** | 11.9 / 32.1 / 35.7 / 34.9 / **36.2** | 18.3 / 29.3 / 34.2 / 36.0 / **37.1** |
| DI | 18.1 / 29.7 / 29.9 / 30.4 / **29.9** | 14.2 / 43.6 / 46.1 / 46.5 / **46.6** | 18.0 / 29.2 / 50.1 / 51.5 / **51.6** |
| SI | 31.0 / 45.1 / 46.1 / 45.1 / **45.8** | 46.7 / 70.9 / 72.0 / 73.4 / **74.0** | 31.0 / 44.6 / 73.0 / 74.3 / **74.7** |
| Admix | 40.2 / 54.9 / 55.5 / 54.9 / **55.5** | 57.0 / 78.0 / 77.6 / 77.9 / **77.6** | 41.4 / 56.0 / 80.0 / 79.9 / **80.8** |
| MI-TI-DI | 50.7 / 55.9 / 57.2 / 56.7 / **56.8** | 41.9 / 74.0 / 79.0 / 81.5 / **82.6** | 50.7 / 56.4 / 77.8 / 80.0 / **81.4** |
| MI-TI-DI-SI | 77.6 / 85.3 / 85.7 / 85.0 / **85.0** | 85.5 / 93.1 / 93.7 / 94.2 / **94.1** | 78.0 / 85.0 / 94.4 / 94.6 / **95.2** |
| MI-TI-DI-Admix | 84.7 / 89.4 / 89.2 / 89.9 / **89.3** | 88.4 / 94.9 / 95.1 / 95.2 / **95.0** | 85.8 / 90.1 / 94.8 / 95.4 / **95.5** |

|  | VGG-16 → ResNet-50 | | |
|---|---|---|---|
|  | Baseline | +RAP | +RAP-LS |
| I | 37.2 / 52.0 / 53.4 / 53.1 / **53.7** | 17.8 / 48.5 / 53.9 / 53.7 / **53.0** | 38.1 / 53.0 / 52.4 / 54.8 / **54.2** |
| MI | 60.2 / 64.3 / 63.5 / 62.0 / **62.5** | 32.9 / 57.1 / 67.6 / 73.1 / **76.2** | 60.4 / 62.0 / 66.3 / 73.2 / **76.4** |
| TI | 45.3 / 62.7 / 63.6 / 62.5 / **62.8** | 19.4 / 56.6 / 63.0 / 65.6 / **64.8** | 46.0 / 62.9 / 63.5 / 65.8 / **65.8** |
| DI | 51.5 / 72.9 / 73.2 / 72.5 / **72.2** | 29.6 / 80.9 / 85.0 / 86.4 / **86.0** | 51.4 / 73.8 / 88.9 / 89.2 / **88.8** |
| SI | 64.6 / 81.0 / 80.2 / 80.5 / **80.0** | 68.1 / 91.9 / 92.3 / 92.4 / **92.7** | 64.9 / 80.6 / 95.1 / 95.3 / **94.7** |
| Admix | 76.8 / 87.5 / 88.2 / 88.0 / **87.3** | 79.4 / 93.8 / 94.4 / 95.2 / **94.6** | 77.6 / 88.3 / 96.6 / 96.8 / **96.8** |
| MI-TI-DI | 81.1 / 89.9 / 89.8 / 90.3 / **90.0** | 66.7 / 94.6 / 96.3 / 96.9 / **97.2** | 81.4 / 88.5 / 96.5 / 97.3 / **97.7** |
| MI-TI-DI-SI | 95.1 / 97.6 / 98.0 / 97.9 / **97.6** | 94.7 / 98.4 / 98.8 / 98.9 / **98.8** | 95.2 / 97.5 / 99.3 / 99.4 / **99.4** |
| MI-TI-DI-Admix | 97.2 / 98.1 / 98.0 / 98.1 / **97.8** | 96.1 / 99.1 / 99.2 / 99.3 / **99.2** | 97.3 / 98.6 / 99.5 / 99.6 / **99.6** |

|  | VGG-16 → DenseNet-121 | | |
|---|---|---|---|
|  | Baseline | +RAP | +RAP-LS |
| I | 35.4 / 50.4 / 49.8 / 48.4 / **49.1** | 15.4 / 46.0 / 49.6 / 50.5 / **50.6** | 35.2 / 50.3 / 49.7 / 52.9 / **51.4** |
| MI | 62.1 / 63.8 / 62.8 / 61.7 / **60.5** | 26.6 / 51.1 / 63.4 / 70.0 / **73.0** | 61.6 / 62.5 / 62.7 / 70.5 / **73.9** |
| TI | 43.5 / 58.6 / 58.7 / 57.2 / **55.9** | 19.4 / 55.8 / 62.7 / 63.0 / **63.7** | 44.3 / 58.3 / 60.3 / 63.8 / **62.1** |
| DI | 48.1 / 70.2 / 68.9 / 70.0 / **68.8** | 26.5 / 79.9 / 82.3 / 84.2 / **85.0** | 47.9 / 70.5 / 85.1 / 87.2 / **87.2** |
| SI | 65.3 / 82.3 / 82.4 / 82.0 / **82.1** | 71.3 / 93.3 / 93.7 / 94.4 / **94.8** | 65.5 / 82.2 / 95.2 / 95.4 / **95.7** |
| Admix | 79.6 / 89.4 / 88.6 / 88.4 / **88.2** | 83.5 / 96.1 / 95.9 / 96.2 / **96.4** | 79.2 / 88.9 / 97.4 / 97.4 / **97.2** |
| MI-TI-DI | 80.3 / 87.0 / 88.7 / 89.3 / **88.8** | 62.9 / 94.0 / 95.9 / 96.4 / **97.0** | 80.4 / 86.8 / 96.8 / 97.2 / **97.3** |
| MI-TI-DI-SI | 95.3 / 98.2 / 98.4 / 98.4 / **98.1** | 95.9 / 99.2 / 99.2 / 99.2 / **99.2** | 95.4 / 98.2 / 99.5 / 99.5 / **99.4** |
| MI-TI-DI-Admix | 97.1 / 98.6 / 98.8 / 99.1 / **98.9** | 97.4 / 99.4 / 99.6 / 99.6 / **99.5** | 97.3 / 98.5 / 99.5 / 99.5 / **99.6** |

|  | Inc-v3 → ResNet-50 | | |
|---|---|---|---|
|  | Baseline | +RAP | +RAP-LS |
| I | 34.0 / 48.4 / 51.2 / 50.1 / **51.5** | 22.7 / 58.6 / 60.9 / 61.1 / **62.1** | 34.5 / 49.0 / 60.2 / 60.5 / **62.0** |
| MI | 58.5 / 59.1 / 60.4 / 60.3 / **62.0** | 43.8 / 77.0 / 81.7 / 84.0 / **85.8** | 58.5 / 59.1 / 80.0 / 82.6 / **84.8** |
| TI | 33.6 / 46.9 / 48.7 / 48.5 / **49.3** | 21.8 / 58.9 / 60.2 / 61.7 / **63.4** | 33.1 / 47.2 / 59.5 / 61.5 / **61.6** |
| DI | 48.4 / 65.8 / 67.2 / 68.4 / **68.4** | 33.3 / 78.8 / 81.4 / 81.4 / **81.7** | 48.3 / 65.7 / 80.7 / 82.3 / **81.8** |
| SI | 43.7 / 61.9 / 63.9 / 65.1 / **66.2** | 45.8 / 67.0 / 69.4 / 69.5 / **69.8** | 43.6 / 62.3 / 72.5 / 73.4 / **72.8** |
| Admix | 56.1 / 73.0 / 75.9 / 76.9 / **75.9** | 57.0 / 77.5 / 79.8 / 80.3 / **80.2** | 56.3 / 73.4 / 82.9 / 84.0 / **84.9** |
| MI-TI-DI | 72.2 / 79.5 / 81.9 / 81.9 / **82.9** | 61.2 / 88.1 / 90.8 / 91.9 / **91.8** | 72.2 / 79.4 / 89.9 / 91.5 / **90.6** |
| MI-TI-DI-SI | 82.9 / 88.3 / 88.3 / 88.4 / **89.0** | 83.5 / 90.8 / 91.2 / 90.6 / **91.2** | 82.8 / 88.1 / 91.9 / 92.6 / **92.3** |
| MI-TI-DI-Admix | 89.8 / 91.6 / 91.3 / 91.4 / **91.5** | 89.0 / 93.9 / 94.0 / 94.0 / **94.1** | 89.6 / 92.3 / 94.1 / 94.8 / **94.7** |

| | Inc-v3 → DenseNet-121 | | |
| --- | --- | --- | --- |
| | Baseline | +RAP | +RAP-LS |
| I | 35.2 / 47.2 / 47.3 / 46.8 / **48.7** | 21.4 / 54.3 / 57.2 / 59.4 / **60.8** | 34.9 / 47.5 / 58.7 / 58.8 / **60.0** |
| MI | 57.4 / 56.2 / 56.5 / 56.8 / **56.7** | 42.9 / 74.0 / 80.1 / 82.5 / **84.6** | 57.4 / 56.2 / 77.4 / 81.9 / **84.6** |
| TI | 35.8 / 48.6 / 47.8 / 48.9 / **49.4** | 22.1 / 59.6 / 63.3 / 65.7 / **63.4** | 35.5 / 48.7 / 61.6 / 64.2 / **63.8** |
| DI | 53.2 / 72.1 / 71.8 / 71.5 / **71.9** | 35.7 / 81.9 / 83.7 / 85.1 / **85.0** | 53.2 / 71.8 / 84.1 / 85.2 / **84.0** |
| SI | 46.6 / 63.7 / 65.1 / 65.9 / **65.9** | 52.6 / 72.4 / 73.5 / 74.5 / **74.9** | 46.6 / 63.0 / 77.7 / 77.9 / **77.2** |
| Admix | 60.5 / 76.7 / 78.0 / 79.3 / **78.5** | 63.9 / 83.2 / 83.4 / 84.1 / **83.7** | 61.9 / 76.9 / 87.7 / 87.3 / **87.4** |
| MI-TI-DI | 76.7 / 84.7 / 85.7 / 85.7 / **85.7** | 65.1 / 91.5 / 92.8 / 94.0 / **94.2** | 76.7 / 84.6 / 92.6 / 92.9 / **93.3** |
| MI-TI-DI-SI | 89.0 / 91.9 / 91.7 / 91.8 / **92.0** | 89.0 / 94.7 / 95.6 / 95.2 / **95.2** | 89.0 / 91.4 / 95.1 / 95.4 / **95.6** |
| MI-TI-DI-Admix | 93.5 / 95.5 / 95.9 / 95.1 / **95.4** | 93.3 / 96.8 / 96.9 / 96.4 / **96.2** | 94.1 / 95.5 / 97.2 / 97.5 / **97.6** |

| | Inc-v3 → VGG-16 | | |
| --- | --- | --- | --- |
| | Baseline | +RAP | +RAP-LS |
| I | 39.9 / 53.1 / 54.1 / 53.7 / **55.1** | 29.1 / 63.0 / 65.8 / 66.9 / **65.9** | 39.7 / 52.6 / 65.6 / 68.3 / **68.0** |
| MI | 60.7 / 62.2 / 63.8 / 62.1 / **63.1** | 50.7 / 76.1 / 81.0 / 83.6 / **84.9** | 60.7 / 62.2 / 79.8 / 84.0 / **84.6** |
| TI | 41.6 / 55.1 / 55.2 / 55.3 / **58.1** | 31.1 / 65.9 / 67.1 / 68.2 / **68.6** | 41.5 / 55.1 / 66.3 / 68.0 / **69.5** |
| DI | 54.9 / 73.4 / 74.5 / 76.0 / **76.1** | 44.4 / 83.4 / 84.7 / 85.0 / **85.2** | 54.9 / 73.0 / 85.7 / 87.2 / **86.4** |
| SI | 46.7 / 62.4 / 64.4 / 65.7 / **66.0** | 47.4 / 67.6 / 69.2 / 68.6 / **69.2** | 46.3 / 64.1 / 72.4 / 72.1 / **73.0** |
| Admix | 57.3 / 73.2 / 72.8 / 74.0 / **74.5** | 57.3 / 75.4 / 75.9 / 77.5 / **77.2** | 55.3 / 73.4 / 82.6 / 82.2 / **83.5** |
| MI-TI-DI | 74.7 / 82.7 / 84.7 / 84.6 / **85.1** | 67.7 / 90.0 / 91.9 / 92.3 / **92.7** | 74.7 / 82.5 / 90.4 / 90.8 / **91.0** |
| MI-TI-DI-SI | 79.8 / 88.0 / 87.6 / 87.5 / **87.6** | 81.6 / 89.0 / 89.4 / 89.4 / **90.3** | 79.7 / 87.8 / 92.4 / 92.5 / **92.9** |
| MI-TI-DI-Admix | 87.9 / 89.7 / 90.7 / 91.4 / **91.4** | 87.0 / 92.2 / 92.3 / 92.5 / **93.2** | 87.7 / 91.7 / 94.5 / 94.6 / **94.1** |