# OpenReview forum: "Boosting the Transferability of Adversarial Attacks with Reverse Adversarial Perturbation"
_NeurIPS.cc/2022/Conference — NeurIPS 2022 Accept_

### Official Review · Reviewer_Ra8U · 2022-06-14

**Rating:** 6
**Confidence:** 5
**Soundness:** 4 excellent
**Presentation:** 3 good
**Contribution:** 3 good

**Summary:**

The authors claim that the adversarial examples should be in a flat local region for better transferability. To boost the transferability, they propose a simple yet effective method named Reverse Adversarial Perturbation (RAP). RAP adds an inner optimization to help the attack escape sharp local minima, which is general to other attacks. Experimental results demonstrate the high effectiveness of RAP.

**Questions:**

See weakness, especially 4 and 5.

**Strengths And Weaknesses:**

# Strengths

1. The idea is novel and interesting. The adversarial example in flat local region could boost the transferability, which seems to be reasonable.

2. The method is simple yet effective. Adding an inner optimization could effectively boost the transferability, which is also general to existing attacks.

3. Extensive evaluations on ImageNet demonstrate the effectiveness of the proposed method in most scenarios.

# Weakness

1. Figure 1 is not clear. What do the x-axis and y-axis indicate? How do you obtain the curve?

2. Why does the flat local region indicate better transferability? It might be interesting if you could provide some analysis.

3. What does one iteration mean? Does it indicate one forward and back propagation?

4. RAP is general to existing attacks, not limited to input transformation based attacks. Hence, the authors should integrate RAP into various attacks, such as momentum-based attack [1], different loss function [2], model specific attack [3].

5. The biggest concern is the huge computation cost due to the larger iteration and the inner optimization in Eq. (4). Existing attacks could exhibit superior attack performance in much lower iterations. For instance, MI-FGSM [4], VMI-FGSM [1], EMI-FGSM [5], and the corresponding input transformations only adopt 10 iterations.

6. Minor typos:

Line 42 “a slightly change…” -> “a slight change…”

Line 65 “help to escape…” -> “help escape…”

Line 76 “can significant boost…” -> “can significantly boost…”


[1] Wang et al. Enhancing the transferability of adversarial attacks through variance tuning. CVPR 2021.

[2] Wu et al. Boosting the Transferability of Adversarial Samples via Attention. CVPR 2020.

[3] Li et al. Learning Transferable Adversarial Examples via Ghost Networks. AAAI 2020.

[4] Dong et al. Boosting Adversarial Attacks with Momentum. CVPR 2018.

[5] Wang et al. Boosting Adversarial Transferability through Enhanced Momentum. BMVC 2021.

---

> ### Author Response · Authors · 2022-08-02
> **Response to Reviewer Ra8U (Part 1/2)**
>
> ### **Q4.1** 'Figure 1 is not clear. What do the x-axis and y-axis indicate? How do you obtain the curve?'
> **Response 4.1:**
> Thanks for pointing out these. Figure 1 is a schematic diagram in 1D space. The x-axis means the value of input $x$. The y-axis means the value of attack loss function $\mathcal{L}$. We will add these annotations in the revision.
>
> ----
>
> ### **Q4.2** 'Why does the flat local region indicate better transferability? It might be interesting if you could provide some analysis.'
>
> **Response 4.2:** Thanks for this insightful suggestion. This is indeed an important question, and the connection between the local flatness and transferability is the main motivation of our method.  In the submitted manuscript, actually, we have analyzed the connection between the flatness of loss landscape and adversarial transferability in **Section 1 (Line 38-43, Line 51-56)** and **Section 3.2 (Line 132-140)**. And we also verified whether RAP could boost the flatness of the adversarial example or not in **Section 3.3** experimentally.  Here we summarize the key points in the following.
>
> 1）The previous works attribute the poor attack transferability to the overfitting of adversarial examples to the surrogate model. In Figure 1, we use a schematic plot to interpret this. As shown in Figure 1 (b), when $x^{pgd}$ locates at a sharp local minimum, it is not stable and is sensitive to changes of $\mathcal{M}^{S}$. When having some changes on model parameters, $x^{pgd}$ could result in a high attack loss against $\mathcal{M}^{S'}$ and lead to a failure transfer attack. In contrast, the flat local minimum is less sensitive to the changes in decision boundary, thus leading to a more stable transfer attack. Following this, we advocate to find $x^{adv}$ located at flat local region and propose to minimize the maximal loss value within a local neighborhood region around the adversarial example $x^{adv}$.
>
> 2）The experimental results demonstrate that RAP can boost the adversarial transferability. We also take loss landscape visualizations to verify whether RAP can help us find a $x^{adv}$ advocated at the local flat region or not. Figure 3 plots the visualizations. We can see that compared to the baselines, the loss landscape of $x^{adv}$ founded by RAP is much flatter.
>
> ----
>
> ### **Q4.3** 'What does one iteration mean? Does it indicate one forward and back propagation?'
> **Response 4.3:**
> We utilize Algorithm 1 of the main submission to clarify this point. For one iteration of inner loop (t= 1,..,T), it indicates one forward and backpropagation. For one iteration of outer loop (k= 1,..., K), it first conducts an inner loop (T iterations) process mentioned above. And after getting $n^{rap}$, it conducts one forward calculation and backpropagation to update $x^{adv}$. We will make this clearer in our next version.

---

> > ### Author Response · Authors · 2022-08-02
> > **Response to Reviewer Ra8U (Part 2/2)**
> >
> > ### **Q4.4** 'RAP is general to existing attacks, ... , different loss function [2], model specific attack [3].'
> >
> > **Response 4.4:**
> > Thanks for this useful suggestion.  According to the reviewer's suggestion, we conduct the evaluation of combining our RAP with the mentioned works. We choose the ResNet-50 as the source model and MI-TI-DI as the baseline method. For the mentioned work [2], we do not find the source code. Therefore, we conduct the experiments for [1] and [3].
> >
> > **(1)** For combining variance-tuning [1] with RAP,  we calculate the variance of $x^{adv}+n^{rap}$ rather than the variance of $x^{adv}$ like the orginal paper. The results are shown in the below table.
> >
> > **Experimental Results:**
> > The targeted attack results (%) are shown in the below table. We also follow the experimental settings in Section 4.1 of the main submission.
> > |targeted attack ($M^S$: Res-50)	| Inc-v3	| Dense-121	| VGG-16|
> > | :----------: | :--------------------------: | :-----------------------------------------: | :------------------: |
> > |MI-TI-DI                    | 10.9 | 74.9 | 62.8|
> > |MI-TI-DI-VT [1]         | 34.4 | 83.4| 75.1|
> > |MI-TI-DI-VT+RAP      | 36.1 | 88.3| 80.3|
> >
> > The experimental results demonstrate that RAP can further boost the adversarial transferability of VT method.
> >
> > **(2)** For combining ghost network [3] with RAP, we conduct the experiments on our PyTorch codes following the original TensorFlow codes provided by the authors. The main idea of ghost network is to perturb skip connections of resnet to generate ensemble networks. To achieve this goal, the authors multiply skip connection by the random scalar r sampled from a uniform distribution.  We reimplement this procedure with the hyperparameter about r  recommended in the orginal paper on the PyTorch resnet-50 model (we utilize the checkpoint from torchvision).
> >
> > **Experimental Results:**
> >
> > The targeted attack results (%) are shown in the below table. We also follow the experimental settings in Section 4.1 of the main submission.  Here, we use GN to represent ghost network method.
> >
> > |targeted attack ($M^S$: Res-50)	| Inc-v3	| Dense-121	| VGG-16|
> > | :----------: | :--------------------------: | :-----------------------------------------: | :------------------: |
> > |MI-TI-DI                    | 10.9 | 74.9 | 62.8|
> > |MI-TI-DI-GN          | 24.9 | 85.9| 80.7|
> > |MI-TI-DI-GN+RAP-LS      | 49.7 | 89.6 | 87.7 |
> >
> > The results show that ghost network method can improve the transfer attack performance. Combined with RAP-LS, the adversarial transferability can be further improved especially on the inception-v3 model.
> >
> >
> > ----
> >
> > ### **Q4.5** 'The biggest concern is the huge computation cost ... input transformations only adopt 10 iterations.'
> >
> > **Response 4.5:** Thanks for this comment. It is indeed that by introducing the inner maximization process, our method will increase the time cost of adversarial example generation process.
> >
> > 1) We firstly analyze the computational cost of our method. In Algorithm 1 with global iteration number $K$, late-start iteration number $K_{LS}$ and inner iteration number $T$, our RAP-LS requires $K+(K-K_{LS})*T$ forward and backward calculation. While the original attack algorithm requires $K$ forward and backward calculation. The extra computation cost of RAP-LS is $(K-K_{LS})*T$ times forward and backward calculation. In our experiments, $T$ is set to 8 and $K_{LS}$ is set to 100. $K$ is set to 400. As for the larger number of outer iteration number $K$, we follow the existing work of [8] to use larger iterations. As pointed out in [8], currently the unreasonably restricting attack optimization to a limited number of iterations leads to poor transferability.
> >
> > 2) However, we would like to emphasize that the adversarial example generation process is conducted based on the offline surrogate models. Compared with this offline time cost, the attacking performance is much more important for black-box attacks, which is also the main advantage of our method. Besides, the late-start strategy could alleviate the time cost. However, we will keep exploring more efficient transfer-based attack methods which could also help to find flat local minimum in our future work.
> >
> > [8]. On Success and Simplicity: A Second Look at Transferable Targeted Attacks, NeurIPS 2021.
> >
> > ----
> >
> > ### **Q4.6** Minor typos:
> >
> > **Response 4.6:** Thanks for pointing out these typos. We have corrected them and thoroughly proofread the revised manuscript.

---

> ### Author Response · Authors · 2022-08-08
> **Discussion Inquiry**
>
> Dear Reviewers,
>
> Thanks for spending time reviewing our work and providing valuable feedback. We have provided new experiments in Response 4.4 as well as clarification in response to your questions and the mentioned weakness part. Do our response address your questions? And do you have other comments?
>
> Best regards,
>
> Authors

---

> ### Author Response · Authors · 2022-08-09
> **Appreciate your further comment and your recognition of our responses**
>
> > **Reviewer Ra8U's further comment**: "Thank the authors for the responses. I think the authors have addressed my main concerns and would maintain my score."
>
> Dear Reviewer Ra8U:
>
> We sincerely appreciate your further comment, and we are greatly encouraged that our responses have addressed your main concerns. All  additional experiments and analysis presented in our first-round responses will be updated into the final version of our manuscript.
>
> Best regards,
>
> Authors

---

### Official Review · Reviewer_HFVj · 2022-07-10

**Rating:** 5
**Confidence:** 4
**Soundness:** 3 good
**Presentation:** 2 fair
**Contribution:** 2 fair

**Summary:**

The paper proposed to add reverse adversarial perturbation at each iteration in gradient-based attacks to compute more accurate gradient directions and escape sharp local minimum

**Questions:**

- How is. the performance of the attack wrt to PGD-based adversarial training

- Recent works of improving gradient directions by looking ahead [a] or by accumulating the gradients of multiple data points around the current data point sampled [b] can be discussed to clearly highlight the contributions

[a] Jang, Donggon, Sanghyeok Son, and Dae-Shik Kim. "Strengthening the Transferability of Adversarial Examples Using Advanced Looking Ahead and Self-CutMix." Proceedings of the IEEE/CVF Conference on Computer Vision and Pattern Recognition. 2022.

[b] Wang, Xiaosen, et al. "Boosting adversarial transferability through enhanced momentum." arXiv preprint arXiv:2103.10609 (2021).

- SOTA transferability technique   Variance tuning-based FGSM [c] as well as [a] should be reported since they refine the gradient direction as in the proposed method. I appreciate the authors for including another line of methods such as LinBP and ILA, and TTP.

[c] Wang, Xiaosen, and Kun He. "Enhancing the transferability of adversarial attacks through variance tuning." Proceedings of the IEEE/CVF Conference on Computer Vision and Pattern Recognition. 2021.

I'm giving the score of 4 because of no evaluation and discussion to  [a,b,c] which also find optimal gradient direction at each step. I'm very much open to increasing the scores given the comprehensive experiments already reported in the paper.

**Ethics Review Area:**

["I don’t know"]

**Limitations:**

-  calculation of inner optimization increases the computational time of attack wrt baseline methods.
- Adds more hyperparameters (S and mu) but the ablation studies show they are largely target model agnostic

**Strengths And Weaknesses:**

#### Strengths
- Paper is clear and the idea is simple.
- Experiments are comprehensive
- Can be integrated easily into existing gradient-based attacks
- Figure 3 is very insightful


#### Weakness
- The paper improves already an earlier line of works to find a more accurate gradient direction. The novelty of the method is not high but that said, the paper clearly provides detailed analyses from a loss landscape perspective and is validated through a battery of experiments in both targeted and untargeted persepctive.

---

> ### Author Response · Authors · 2022-08-02
> **Response to Reviewer HFVj (Part 1/2)**
>
> ### **Q3.1** 'How is. the performance of the attack wrt to PGD-based adversarial training'
> **Response 3.1:**
> Thanks for this kind suggestion. Actually, we have tested transfer attacks on PGD-based adversarial training in Line 276-280 on Page 8. And the detailed results are shown in Section C and Table 8-9 of the appendix due to space limitation.
>
> We conducted experiments on two multi-step PGD Adversarial training (AT) models from [1] (the $\ell_{\infty}$ ResNet-50 AT model with budget $4/255$ that ranks first in the RobustBench leaderboard [2] and the $\ell_{2}$ ResNet-50 AT model with budget 0.5). We also tested our attacks on the stronger defense models, the combination of PGD-AT models and NRP, an adversarial purified defense method. The results are shown in Table 8 and 9 of the appendix. We can observe that our RAP further boosts the transferability of baseline methods on PGD-AT models.
>
> [4] . https://github.com/microsoft/robust-models-transfer
>
> [5] . https://robustbench.github.io/
>
> ----
>
> ### **Q3.2** 'Recent works of improving gradient directions by looking ahead ... can be discussed to clearly highlight the contribution'
>
> **Response 3.2:**
> Thanks for this kind suggestion. Our RAP and the works mentioned by the reviewer have a similar goal, to find a more stable gradient direction to escape sharp local minimum. Being different from RAP, these methods randomly sample points around the current $x^{adv}$ to calculate the more stable gradient. In contrast, to find $x^{adv}$ located at flat local region, our RAP minimizes the maximal loss value within a local neighborhood region around the current $x^{adv}$.
> According to the reviewer's suggestion, we conduct the evaluation of our RAP and the mentioned works. We choose the ResNet-50 as the source model and MI-TI-DI as the baseline method. [a] is a recent CVPR 2022 paper and we do not find the source code.
> Experimental Results:
> The untargeted and targeted attack results (%) are shown in the below tables. We also follow the experimental settings in Section 4.1 of the main submission.
> |untargeted attack ($M^S$: Res-50)	| Inc-v3	| Dense-121	| VGG-16|
> | :----------: | :--------------------------: | :-----------------------------------------: | :------------------: |
> |MI-TI-DI                    | 85.7 | 99.8 | 99.8|
> |MI-TI-DI-VT [b]         | 95.8 | 100| 100.0|
> |EMI-TI-DI [c]            | 93.6| 100.0 | 100|
> |MI-TI-DI+RAP-LS    | 96.9| 100.0| 100|
>
> |targeted attack ($M^S$: Res-50)	| Inc-v3	| Dense-121	| VGG-16|
> | :----------: | :--------------------------: | :-----------------------------------------: | :------------------: |
> |MI-TI-DI                    | 10.9 | 74.9 | 62.8|
> |MI-TI-DI-VT [b]         | 34.4 | 83.4| 75.1|
> |EMI-TI-DI [c]            | 32.1| 81.2 | 73.3|
> |MI-TI-DI+RAP-LS    | 33.2| 88.5| 81.5|
>
> As shown in experimental results, our RAP-LS achieves better performance, especially for targeted attacks. Compared with VT [b], RAP-LS gets an increase of 3.4% for targeted attacks in terms of average success rate. This demonstrates the effectiveness of our methods. We will add these in the revision.

---

> > ### Author Response · Authors · 2022-08-02
> > **Response to Reviewer HFVj (Part 2/2)**
> >
> > ### **Q3.3** 'calculation of inner optimization increases the computational time of attack wrt baseline methods.'
> > **Response 3.3:**
> > Thanks for this comment. It is indeed that by introducing the inner maximization process, our method will increase the time cost of the adversarial example generation process.
> >
> > 1) We firstly analyze the computational cost of our method. In Algorithm 1 with global iteration number $K$, late-start iteration number $K_{LS}$ and inner iteration number $T$, our RAP-LS requires $K+(K-K_{LS})*T$ forward and backward calculation. While the original attack algorithm requires $K$ forward and backward calculation. The extra computation cost of RAP-LS is $(K-K_{LS})*T$ times forward and backward calculation. In our experiments, $T$ is set to 8 and $K_{LS}$ is set to 100. $K$ is set to 400. As for the larger number of outer iteration number K, we follow the existing work of [8] to use larger iterations. As pointed out in [8], currently the unreasonably restricting attack optimization to a limited number of iterations leads to poor transferability.
> >
> > 2) However, we would like to emphasize that the adversarial example generation process is conducted based on the offline surrogate models. Compared with this offline time cost, the attacking performance is much more important for black-box attacks, which is also the main advantage of our method. Besides, the late-start strategy could alleviate the time cost. However, we will keep exploring more efficient transfer-based attack methods which could also help to find flat local minimum in our future work.
> >
> > [8]. On Success and Simplicity: A Second Look at Transferable Targeted Attacks, NeurIPS 2021.
> >
> > ----
> >
> > ### **Q3.4** 'Adds more hyperparameters (S and mu) but the ablation studies show they are largely target model agnostic'
> > **Response 3.4:**
> > We do not contain the parameter $S$ and $mu$ mentioned by the reviewer. It would be greatly appreciated if the reviewer can provide further comments to help us to understand more correctly. The ablation study of the parameters of RAP is shown in Section 4.6.

---

> > > ### Comment · Reviewer_HFVj · 2022-08-09
> > > **Thank you for the response**
> > >
> > > Dear Authors,
> > >
> > > I went through your response, other reviews, and the paper again.
> > >
> > > My concerns are partially addressed as the recent competitors [b, BMVC2021] and [c, CVPR21] are not reported in the main paper and provided results for Res50 are not sufficient.  Nonetheless, I strongly suggest adding the results of [b, BMVC2021] and [c, CVPR21] to the revised main paper and hope the authors include the complete set of experiments in the revised version.  I increase my score to 5  taking into account the novelty and empirical evidence.
> > >
> > > Moreover, the LinBP method is especially focused on skip connections-based networks and not generic. Therefore,  I urge you to move the results of LinBP to supplementary and add competitive baselines [b, c, and a if possible] in Table 5 of the main paper.
> > >
> > > I apologize for my comment on S and mu. It was inadvertently added here from a different paper. Please ignore.

---

> ### Author Response · Authors · 2022-08-08
> **Discussion Inquiry**
>
> Dear Reviewers,
>
> Thanks for spending time reviewing our work and providing valuable feedback. We have provided new experiments in Response 3.2 as well as clarification in response to your questions and the mentioned limitations. Do our response address your questions? And do you have other comments?
>
> Best regards,
>
> Authors

---

> > ### Comment · Reviewer_Ra8U · 2022-08-09
> > **Thanks for the responses**
> >
> > Thank the authors for the responses. I think the authors have addressed my main concerns and would maintain my score.

---

> ### Author Response · Authors · 2022-08-09
> **Greatly encouraged by your recognition of our responses and the increased score**
>
> > **Reviewer HFVj's further comments**: "I went through your response, other reviews, and the paper again. My concerns are partially addressed as the recent competitors [b, BMVC2021] and [c, CVPR21] are not reported in the main paper and provided results for Res50 are not sufficient. Nonetheless, I strongly suggest adding the results of [b, BMVC2021] and [c, CVPR21] to the revised main paper and hope the authors include the complete set of experiments in the revised version. I increase my score to 5 taking into account the novelty and empirical evidence..."
>
> Dear Reviewer HFVj:
>
> We sincerely appreciate the recognition of our novelty and empirical evidence. And, we are greatly encouraged that our additional results help to address your concerns. We will add the complete set of experimental comparisons to [b] and [c] in our revision, as follows:
>
> - In addition to the experimental results reported in our first-round responses (where we adopted ResNet-50 as the surrogate model and Inc-v3/Dense-121/VGG-16 as target models), we will complete the experiments by taking Inc-v3/Dense-121/VGG-16 as surrogate models, respectively.
>
> Besides, thanks for your constructive suggestion about LinBP. We will follow the suggestion to move LinBP to supplementary and add the comparisons to [b,c] in the main manuscript. We will also add the comparison to [a] once we got their source code.
>
> Thanks again for your precious time and constructive comments, which are really beneficial for us to improve this work and highlight its contributions.
>
> Best regards,
>
> Authors

---

### Official Review · Reviewer_pUE8 · 2022-07-11

**Rating:** 5
**Confidence:** 3
**Soundness:** 3 good
**Presentation:** 3 good
**Contribution:** 2 fair

**Summary:**

This paper focuses on generating transferable adversarial examples. Observing previous works overfits to the surrogate model, the authors propose a novel method to address this issue. According to the experiments, the proposed method generates stronger transferable adversarial examples.


**Questions:**

See weakness. If the concern in weakness can be well addressed, I am willing to increase the score.

**Limitations:**

1. It is not clear if the proposed method can attack strong defense models.
2. It is not clear if the proposed method can be applied to generate other kinds of adversarial examples (such as patch attack)

**Strengths And Weaknesses:**

+ Strength
1. The proposed method is interesting and insightful. Previous methods usually address the overfitting issues by borrowing the techniques from deep learning model training strategy, such as momentum term, data augmentation. In contrast, this paper proposes a method specifically designed for generating adversarial examples.
2. The results are reasonably good. According to the results in the main paper, the proposed RAP and RAP-LS significently improve the transferability of adversarial examples.
- Weakness
I found the proposed method is less effective when attacking strong defense models (such as Feature Denoising), according to Table 8. Is there any explanation?

---

> ### Author Response · Authors · 2022-08-02
> **Response to Reviewer pUE8**
>
> ### **Q2.1** 'I found the proposed method is less effective when attacking strong defense models (such as Feature Denoising), according to Table 8. Is there any explanation?'
>
> **Response 2.1**:
> Thanks for this insightful suggestion. We think this is mainly due to **the specially designed feature denoising block** (i.e., the non-local block), and **the different settings of maximum perturbation size during adversarial training**, as follows.
>
> 1) Feature Denoising [33] inserts several non-local blocks into network to eliminate the adversarial noise at the feature level. According to [33], for input feature map $F_{i}$, the non-local block computes a denoised output feature map $F_{o}$ by taking a weighted average of input features in all spatial locations. Through this, the non-local block would model the global relationship between features in all spatial locations, which may smooth the learned decision boundary. Recalling that our RAP is to boost the transferability by seeking for a flat local minimum. The smoothness of decision boundary could make it harder to escape from certain local minimum, especially for small attack perturbation size, so as to limit the performance improvement of RAP.
>
> 2) In our experiment, for Feature Denoising, the maximum perturbation size during their training is set to 16/255. In Table 8, the maximum perturbation size of attack is also set to 16/255. The attack size of 16/255 may not be large enough for escaping from local minima for the Feature Denoising model trained with 16/255 perturbation size. In contrast, the maximum perturbation size of AT-$\infty$ during training is 4/255. To verify this, we conduct an ablation study of increasing the maximum perturbation size to 20/255. Using a larger perturbation size of 20/255, the attacking performance against Feature Denoising is 48.3% for MI-TI-DI and 50.7% for MI-TI-DI+RAP-LS. The relative performance improvement of RAP-LS is 2.4%, which is much larger than the relative performance improvement of 0.3% in Table 8 with perturbation size 16/255, which may partially explain the phenomenon.
>
> We will add above analysis to the revised manuscript and appendix.
>
> [33] Feature denoising for improving adversarial robustness, CVPR 2019.
>
> ----
>
> ### **Q2.2** 'It is not clear if the proposed method can attack strong defense models.'
> **Response 2.2:**
> We have tested several strong defense models, which we mentioned in **Line 276-280 of Page 8 and the detailed results are shown in Section C and Table 8-10 of the appendix due to space limitation**. To be specific, apart from the Feature Denoising defense method, we also tested two multi-step PGD Adversarial training (AT) models [1] (the $\ell_{\infty}$ ResNet-50 AT model with budget $4/255$ that ranks first in the RobustBench leaderboard [2] and the $\ell_{2}$ ResNet-50 AT model with budget 0.5), a adversarial purified defense method (NRP) [2], and one input transformation method (R\&P) [3]. We also tested our attacks on the stronger defense, the combination of AT models and NRP. We can observe that our RAP further boosts the transferability of baseline methods on these strong defense models. We refer to the appendix for the detailed experimental results.
>
> [1]  https://github.com/microsoft/robust-models-transfer
>
> [2]  A self-supervised approach for adversarial robustness, CVPR 2020.
>
> [3] Mitigating adversarial effects through randomization, ICLR 2018.
>
> ----
>
>
> ### **Q2.3** 'It is not clear if the proposed method can be applied to generate other kinds of adversarial examples (such as patch attack)'
>
> **Response: 2.3**
> Thanks for this constructive comment. As formulated in Eq. (2), the only change of our RAP method is inserting one reverse adversarial perturbation $\mathbf{n}^{rap}$ onto the adversarial example $\mathbf{x}^{adv}$, based on the original objective function of generating adversarial example (*i.e.*, Eq. (1)). Thus, currently we don't see technique obstacles for applying our RAP method with other kind of adversarial examples. Take the mentioned patch attack as an example, if we didn't misunderstand its meaning, we can introduce a binary mask into Eq. (2), *i.e.*, replacing $\mathbf{x}^{adv} + \mathbf{n}^{rap}$ by $(\mathbf{x}^{adv} + \mathbf{n}^{rap}) \cdot \mathbf{m}$. Due to the limited rebuttal period, we have not implement ed it. We will keep the study of combining our RAP method with more types of adversarial examples to explore more advantages of RAP in our future work.

---

> ### Author Response · Authors · 2022-08-08
> **Discussion Inquiry**
>
> Dear Reviewers,
>
> Thanks for spending time reviewing our work and providing valuable feedback. We have provided new response to your questions and mentioned limitations. Do our response address your questions? And do you have other comments?
>
> Best regards,
>
> Authors

---

> > ### Comment · Reviewer_pUE8 · 2022-08-08
> > **Thanks for the response**
> >
> > Hello Authors,
> >
> > I go through the paper, your response, and other reviewers' comments. I think most of my concerns are addressed. The only one concern is: it seems the proposed method is less helpful when the decision boundary is smooth, and current and future models tend to have more smooth decision boundary (eg, using SiLU activation function instead of ReLU). Therefore, I am worried about if the proposed method can be used on future models.
> >
> > I have no other concerns -- the motivation of this paper is clear, and all results make sense to me. I am going to keep Borderline accept unless the authors have more comments.

---

> > > ### Author Response · Authors · 2022-08-08
> > > **Thanks for this insightful comment, but no necessary to overly worry about the effect of our method in future models**
> > >
> > > Dear Reviewer pUE8,
> > >
> > > We greatly appreciate your further response and insightful comment. We are also encouraged by your positive comments on our motivation and results.
> > >
> > > In terms of your remaining concern, "it seems the proposed method is less helpful when the decision boundary is smooth, and current and future models tend to have more smooth decision boundary (eg, using SiLU activation function instead of ReLU)", unfortunately there is no time left for us to further experimentally verify it. However, we would like to share our thoughts according to the current results and our experiences:
> > > - **Is there a tend to use SiLU instead of ReLU in current and future models?** We are not the expert of designing model architectures, thus we cannot sure that whether there is really a tend to replace ReLU by SiLU.  However,  according to the demonstration in the implementation of SiLU in Pytorch, SiLU was originally proposed in 2016 (https://arxiv.org/pdf/1606.08415.pdf). We can see that the standard ReLU is still widely used in the mainstream model architectures, possibly due to its simplicity and low computational cost. In contrast, the sigmoid function in SiLU will lead to high computational cost, which may restrict its usage in the scenario where the computational resource is limited (e.g., deploying the deep model in phone or drone). Maybe SiLU could bring in some benefits to the model, but it cannot fully replace the standard ReLU, as they have different advantages. Besides, our finding in this work that the model with smoother loss landscape is more resistant to transfer attack provides another reason to choose SiLU, which is also one contribution of our work to the community.
> > > - **How smooth of the decision boundary if replacing ReLU by SiLU? Is it really resistant enough to transfer attack?**  As claimed above, unfortunately there is no time left for us to further experimentally verify it. But, we don't think that there is one once-and-for-all solution to make the model's loss landscape smooth enough to be resistant to query-based or transfer-based adversarial attacks. If so, then we have found a very good solution to improve the adversarial robustness. The technique like SiLU may improve the overall smoothness, but it cannot guarantee the enough smoothness of any local region, thus there is still space for improving the adversarial transferability using our RAP method. Besides, note that we have evaluated our method on attacking against the adversarially trained (AT) model (see Section 4.5 and Line 276-286), which is recognized as one of the most effective approach to improve the adversarial robustness, possibly due to that it could enhance the smoothness of loss landscape (please refer to "Semi-supervised robust training with generalized perturbed neighborhood", Pattern Recognition 2022). It is shown that our RAP method still shows very significant improvement on transfer attacks against these AT models (over 10\% improvement, see Line 285-286). We believe that our RAP method will be helpful even for attacking against model with smoother loss landscape.
> > >
> > > **In summary**, we think there is no necessary to overly worry about that our RAP method may not very helpful for future models. Hope you can understand that due to the limited time, our above descriptions may be not fully exact in all sentences, but we think the main points should have been clearly demonstrated. Hope our quick responses could help you to re-evaluate the contribution of our work, including the insight, effectiveness, and the property of plug-and-play (please refer to the response to Q1.1 of Reviewer VMp1). We sincerely thank the reviewer's insightful comment and help again, and the discussion with you is very delightful and beneficial. We are glad to provide more detailed responses to any further concern, if there is still rebuttal time.
> > >
> > > Sincerely,
> > > Authors

---

> > > ### Author Response · Authors · 2022-08-10
> > > **Looking forward to your further feedback about our latest response**
> > >
> > > Dear Reviewer pUE8,
> > >
> > > Considering the discussion stage is close to the end, we are looking forward to your further feedback about our latest response (posted one day ago, see below), whether your remaining concern has been addressed. We would like to discuss with you in more details.
> > > Greatly appreciate your help.
> > >
> > > Sincerely,
> > >
> > > Authors

---

### Official Review · Reviewer_VMp1 · 2022-07-13

**Rating:** 5
**Confidence:** 4
**Ethics Flag:** Yes
**Soundness:** 3 good
**Presentation:** 3 good
**Contribution:** 2 fair

**Summary:**

In this paper, the authors propose a method (RAP) for generating adversarial examples that are transferable to other models. The authors identify that adversarial examples lay on a flat minimum will be more likely to be transferred to other models. The authors then propose the reverse adversarial perturbation guided by an additional inner maximization optimization process, to generate adversarial examples lay on a flat minimum.

**Questions:**

Though It seems that the theory of flat minimum is intuitive, is there any additional analysis besides the empirical experimental results that justify the theory? Additionally, why simple EOT method is not enough for generating adversarial examples at a flat minimum? It seems to the reviewer that no competitive baselines are provided in the paper.

**Ethics Review Area:**

["Privacy and Security (e.g., consent)"]

**Limitations:**

The limited novelty of the paper. The theory of flat minimum is intuitive and not well justified by either experiment or theoretical proofs. Competitive baselines are not provided to show the effectiveness of the proposed method.

**Strengths And Weaknesses:**

+ The proposed method is straightforward and effective.
+ The authors conduct a comprehensive evaluation of several models with various types of attacks.
+ The authors conduct a case study with Google Cloud Vision API to validate the effectiveness of the proposed method.
- The proposed method is not new.
- The authors did not provide an analysis or theoretical proof for justifying the flat minimum hypothesis.
- The evaluation does not have competitive baselines.

---

> ### Author Response · Authors · 2022-08-02
> **Response to Reviewer VMp1**
>
> ### **Q1.1** 'The proposed method is not new.'
>
> **Response 1.1:** Thanks for your comment. Although measuring one work's novelty is somewhat subjective, in the following we would like to summarize our main contributions to clarify our novelty.
>
> **(1) We provide a novel perspective to explain the adversarial transferability** that adversarial examples located at flat local minimum will be more transferable than those at sharp local minimum. We also provide some intuitive analysis and experimental verification about this perspective (see more details in the response to Q1.2). In contrast, we find that in many existing transfer-based attack works, the motivation was just addressing the overfitting to surrogate models, and borrowed "the techniques from deep learning model training strategy, such as momentum term, data augmentation" (**please refer to the first comment by Reviewer pUE8**), without deep analysis.
>
> **(2)** We propose an **insightful**, **effective**, and **plug-and-play** method. **(a)** "**Insightful**" means that our method is inspired by the above perspective, aiming to find adversarial examples located at flat local minimum, which is intuitively illustrated (see Figure 1) and experimentally verified (see Figure 3). **(b)**"**Effective**" means that the proposed RAP method achieves superior attack performance, which is supported by extensive experiments. **(c)** "**plug-and-play**" means that our RAP method can be naturally combined with many existing transfer-based attack method, as RAP focused on replacing the random perturbation to attacked sample by a reverse adversarial perturbation, which has never been studied in exsiting methods. The experimental results also demonstrate that our RAP method could improve attack performance of all existing methods.
>
> ----
> ### **Q1.2** 'Though It seems that the theory of flat minimum is intuitive, is there any additional analysis besides the empirical experimental results that justify the theory? '
>
> **Response 1.2:** Thanks for this valuable comment. In addition to the experimental verification of the connection between the flat minimum and the adversarial transferability, we also provide the following analysis.
>
> 1) As shown in Figure 1, we provided schematic plots to help understand the difference between sharp and flat local minima, as well as their different effects on transferability.  As analyzed in Line 38-43, Line 51-56, Figure 1 intuitively illustrates that the flat local minima is more stable to the change of model parameters (i.e., loss landscape).
>
> 2) In Section 3.3, we take loss landscape visualizations to verify that RAP can help to find adversarial example $x^{adv}$ located at the local flat region. As shown in Figure 3, compared to the baselines, the loss landscape of $x^{adv}$ founded by RAP is much flatter.
>
> We will keep studying the theoretical connection between transferability and flatness of loss landscape in our future work.
>
> ----
>
> ### **Q1.3** 'Additionally, why simple EOT ... competitive baselines are provided in the paper.'
>
>
> **Response 1.3:** Thanks for this kind suggestion. We conducted the experiment of the suggested EOT baseline. We choose the ResNet-50 as the source model and MI-TI-DI as the baseline method. Instead of adding input transformation once like DI, we sample random transformation (resizing and padding) multiple times in each iteration. Then, we add them to $x^{adv}$ following the expectation of transformation (EOT). We set the number of sampling as $10$. Our RAP can be also naturally combined with EOT attack. Therefore, we also take experiments for this combination.
>
> **Experimental Results:**
> The targeted attack results (%) are shown in the below table. We also follow the experimental settings in Section 4.1 of the main submission.
> |Attack ($M^S$: Res-50)	| Inc-v3	| Dense-121	| VGG-16|
> | :----------: | :--------------------------: | :-----------------------------------------: | :------------------: |
> |MI-TI-DI                     | 10.9 | 74.9 | 62.8|
> |MI-TI-DI-EOT             | 11.2 | 76.9| 66.9|
> |MI-TI-DI+RAP            | 28.3| 78.2 | 72.9|
> |MI-TI-DI-EOT+RAP    | 32.8| 86.1| 79.5|
>
> As shown in the above table, **1)** the EOT attack gets a moderate increase on attack performance compared with the baseline MI-TI-DI attack, which demonstrates that EOT could improve adversarial transferability. **2)** Our RAP attack achieves better performance and surpasses the EOT attack by a large margin, especially for Inc-v3 and VGG-16 target models. **3)** Combining RAP with EOT can further boost EOT attack performance.
>
> These results demonstrate that RAP could achieve better adversarial transferability and help find better flat local minima. Besides, the combination of RAP and EOT achieves the best performance among them, which demonstrates that these two methods could complement each other. We will add this part in the revision.

---

> ### Author Response · Authors · 2022-08-08
> **Discussion Inquiry**
>
> Dear Reviewers,
>
> Thanks for spending time reviewing our work and providing valuable feedback. We have provided new experiments in Response 1.3 as well as clarification in response to your questions and the mentioned weakness part. Do our response address your questions? And do you have other comments?
>
> Best regards,
>
> Authors

---

### Review · Ethics_Reviewer_SSxv · 2022-08-08

**Recommendation:**

Most directly the authors can discuss potential mitigations against their work. Experimentally,
they can analyze a larger set of defenses and comment on which of them is most resilient to their
proposed methods (and hopefully discuss why). Presently the paper experiments with only 2 defenses
that I can tell which may offer only a limited basis for extrapolating mitigation recommendations.


**Ethical Issues:**

Yes

**Ethics Review:**

Paper discusses improvements to adversarial attacks against machine learning models. It offers no
discussion or recommendations regarding defending against these improvements and thus falls on the
malicious side of the power balance noted in the NeurIPS ethical guidelines:

  "On the methodology side, for example, a new adversarial attack might give unbalanced power to
   malicious entities; in this case, defenses and other mitigation strategies would be expected, as
   is standard in computer security."

---

### Author Response · Authors · 2022-08-09
**Common response to all reviewers and area chairs**

Dear Reviewers and Area Chairs:

We sincerely appreciate all of your precious time and constructive comments.

We are greatly encouraged by the positive comments of our work, including **the idea is novel, interesting and insightful** (by pUE8, Ra8U); **the proposed method is effective** (by VMp1, pUE8 and Ra8U); **the results are reasonably good** (by pUE8 and Ra8U); **the experiments are comprehensive** (by VMp1, HFVj and Ra8U); **can be integrated easily into existing gradient-based attacks** (by HFVj) and **is general to existing attacks** (by Ra8U); **conducting a case study with Google Cloud Vision API** (by VMp1), etc.

We are also glad to see that **our responses are helpful for addressing the reviewers' concerns**. All these concerns are very insightful and beneficial for us to improve the quality of this work. We will carefully revise our manuscript by adding all suggested experimental comparisons, adding more detailed explanations and fixing the typos, as promised in our first-round responses.

Finally, we would like to emphasize again that the adversarial transferability plays an important role in adversarial machine learning and its intrinsic mechanism is still uncovered. We are confident that our efforts of **providing a novel perspective to explain the adversarial transferability** and **proposing an insightful, effective, and plug-and-play method for transfer adversarial attack** have shed some light on this interesting topic, and could inspire more research on this topic from the community.

Best regards,

Authors

---

### Meta-Review · Area_Chair_Fz4L · 2022-08-27

**Recommendation:** Accept
**Confidence:** Less certain

**Metareview:**

This paper studies the transferability of adversarial examples. In general, the reviewers found the paper is well motivated, and the proposed method is simple and effective. Most initial concerns were about missing comparisons and ablations.

All these concerns are well addressed in the rebuttal. As a result, all reviewers unanimously agree to accept this submission.

**Award:**

No

---

### Decision · Program_Chairs · 2022-09-14

Accept